# Attention as Inference via Fenchel Duality

## Abstract

Attention has been widely adopted in many state-of-the-art deep learning models. While the significant performance improvements it brings have attracted great interest, attention is still poorly understood theoretically. This paper presents a new perspective to understand attention by showing that it can be seen as a solver of a family of estimation problems. In particular, we describe a convex optimization problem that arises in a family of estimation tasks commonly appearing in the design of deep learning models. Rather than directly solving the convex optimization problem, we solve its Fenchel dual and derive a closed-form approximation of the optimal solution. Remarkably, the solution gives a generalized attention structure, and its special case is equivalent to the popular dot-product attention adopted in transformer networks. We show that T5 transformer has implicitly adopted the general form of the solution by demonstrating that this expression unifies the word mask and the positional encoding functions. Finally, we discuss how the proposed attention structures can be integrated in practical models and show that the convex optimization problem indeed provides a principle justifying the attention module design.

## 1 Introduction

Attention-based deep neural networks are now integrated into cutting-edge language models that have revolutionized a broad range of tasks: machine translation (Bahdanau et al., 2014; Luong et al., 2015), sentiment classification (Wang et al., 2016), image captioning (Xu et al., 2015) and unsupervised representation learning (Devlin et al., 2019), etc. Especially, attention plays a pivotal role in the construction of the transformer architecture (Vaswani et al., 2017), which has had a profound impact on the deep learning field.

Despite great empirical success, the design principle of attention has not been well studied in the literature, and there is no in-depth understanding as to why attention-based models (e.g. BERT (Devlin et al., 2019)) have significantly better performance than other models. This lack of understanding impedes practitioners from using attention layers confidently and appropriately, making it challenging to develop new attention-based neural architectures.

In this paper, we offer a new perspective for understanding attention by showing that it is in fact a solver for a certain type of optimization problem that corresponds to an inference task. We give several examples, all of which can be characterized as follows: given 1) an unreliable estimate of the mean of an unknown distribution $p$ on $\mathbb{R}^d$ and 2) a preference distribution $u$ on $\mathbb{R}^d$ encoding beliefs on $p$'s selection, the inference task is to get a better estimate of $p$'s mean given its unreliable estimate and $u$. We derive a convex optimization problem that is abstracted from the task and solve it by instead solving its Fenchel dual (Rockafellar, 1970, p.104). Remarkably, the derived expression of the improved estimate of $p$ gives a generalized attention structure whose special case is equivalent to the popular dot-product attention (Luong et al., 2015) that is also applied in the transformer network (Vaswani et al., 2017). In addition, we show that our generalized attention expression has been implicitly adopted by T5 transformer (Raffel et al., 2020) as the expression unifies the concept of word masks and its positional encoding functions. Extra examples are given to show how the generalized attention structures can be used in practice, and a novel optimal transport (OT)-based attention is derived to show how our framework helps develop more general attention structures. Additionally, experiments are performed, which validates our theoretical work.

## 2 Related work

Since 2019, several authors have investigated the properties and working mechanism of attention. This series of works mainly addresses whether the attention mechanism can serve as a proxy of saliency (Michel et al., 2019; Voita et al., 2019; Jain & Wallace, 2019; Wiegreffe & Pinter, 2019; Serrano & Smith, 2020; Vashishth et al., 2020). Most of these works obtain insights into the attention mechanism by performing empirical studies. The related methods include analyzing the behaviours of trained attention-based models (Clark et al., 2019), pruning a few heads, analyzing the effects of altering the attention weights (Michel et al., 2019; Voita et al., 2019), or a mixture of these (Jain & Wallace, 2019; Vashishth et al., 2020).

Apart from understanding attention empirically, some theoretical results presented by Brunner et al. (2019) and Hahn (2020) show that the self-attention layers are not identifiable. This implies there could exist multiple combinations of attention weights that can provide equally good final predictions. In particular, such non-uniqueness means that the use of attention may complicate interpretability. Besides, Tsai et al. (2019) present a new formulation of attention via the lens of kernels and show that attention can be seen as applying kernel smoother over the inputs. Another important approach to understand attention is to analyze its asymptotic behaviour when the number of heads and the network width approach infinity (Yang, 2019; Hron et al., 2020). In this limiting case, the entire network can be seen as a Gaussian process (Lee et al., 2018) and its behaviours can be characterized by closed-form expressions that are not available in the finite regime.

Very recently (since 2021) several theoretical works have appeared that study attention outside the asymptotic regime. Lu et al. (2021) set up a simple attention-based classification model and derive a closed-form relationship between the word's embedding norm and the product of its key and the query. They empirically show that such relationship also exists in a more complicated and practical configuration. Ramsauer et al. (2021) construct an equivalence relationship between attention and a newly proposed Hopfield network with continuous states. In particular, they show that the new Hopfield network's update rule is equivalent to the attention mechanism used in transformers (Vaswani et al., 2017).

## 3 Setup of a design problem

Throughout the rest of the paper, we consider a prediction task: given an input $\mathbf{X}$, predict an output quantity $\mathbf{Y} = (Y^{(1)}, Y^{(2)}, \ldots, Y^{(K)})$, which includes $K$ components. To be more concrete, we present a few machine learning problems and let them run through our development. We will show that the problems can be unified and abstracted in a similar way and be solved using a generalized attention structure.

**Translation Problem (TP).** In this problem, the input $\mathbf{X}$ is a sentence, or a sequence of words, in the source language. Output $\mathbf{Y}$ is the sequence of words in the target sentence, where $Y^{(k)}$ is the $k^{\text{th}}$ word.

**Image Captioning (IC).** In this problem, the input $\mathbf{X}$ is a raw image and output $\mathbf{Y}$ is the sequence of words in the caption, where $Y^{(k)}$ is the $k^{\text{th}}$ word.

**Filling in the Blanks Task (FB)**. This task has been used to train the BERT model (Devlin et al., 2019). The input $\mathbf{X}$ is a sequence of words with certain percentage of words masked. The output $\mathbf{Y}$ are the predicted masked words, where $Y^{(k)}$ denotes the $k^{\text{th}}$ masked one.

The objective of any of these problems and that we address in this paper is to learn a function $\mathcal{F}$, mapping from the space of $\mathbf{X}$ to the space of $\mathbf{Y}$ so that $\mathbf{Y} = \mathcal{F}(\mathbf{X})$. We will denote by $F^{(k)}$ the part of $\mathcal{F}$ responsible for predicting $Y^{(k)}$ (Fig 1a), namely, $Y^{(k)} = F^{(k)}(X)$. Although we here express $\mathcal{F}$ as separate functions $(F^{(1)}, F^{(2)}, \ldots, F^{(K)})$, we note that it is in fact possible that different $F^{(k)}$'s share some component in common. Without loss of generality, we now focus on the design of $F^{(k)}$.

### 3.1 The Design Problem

In deep learning research, a typical approach to solve the three running tasks is first to use some choice of neural network to extract vector representations $\{\mathbf{t}_1^{(k)}, \mathbf{t}_2^{(k)}, \ldots, \mathbf{t}_M^{(k)}\} \subseteq \mathbb{R}^d$ of $\mathbf{X}$, which are referred as

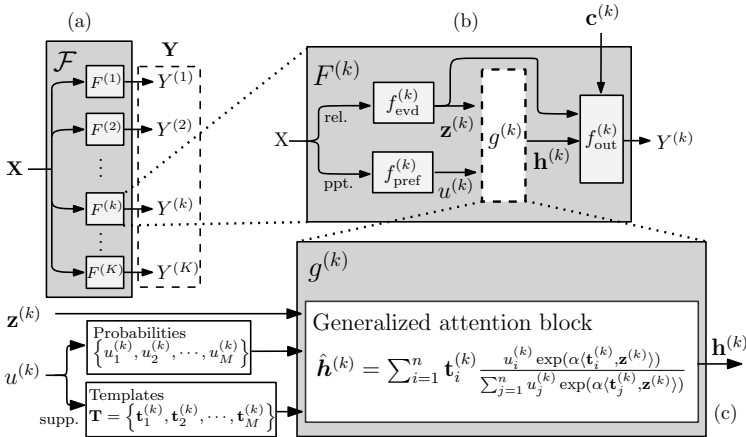

Figure 1: A conceptual graph of the deep learning model that we work with. The block $g^{(k)}$ is the one we will investigate. (a) plots the general structure of a sequence generation model, where block $F^{(k)}$ is responsible for its $k$-th output. This paper focuses on $F^{(k)}$ with the architecture presented in (b) that contains component $g^{(k)}$ inferring a distribution's mean $\mathbf{h}^{(k)}$ based on its noisy estimations from two aspects: its preference (prior) distribution $u^{(k)}$ and a noisy estimation of its mean shift $\mathbf{z}^{(k)}$ from $u^{(k)}$'s. We will show that $g^{(k)}$ should implement the expression presented in (c) whose special case is the familiar dot-product attention (Luong et al., 2015).

templates. Collectively, we will denote the set $\{\mathbf{t}_1^{(k)}, \mathbf{t}_2^{(k)}, \ldots, \mathbf{t}_M^{(k)}\}$ of templates by $\mathbf{T}^{(k)}$.[1] (If $\mathbf{X}$ are words, typical choices of neural network include RNN, LSTM, etc. If $\mathbf{X}$ is an image, a typical choice is CNN.) Let $\mathcal{A} \subseteq \mathbb{R}^d$ denote the space containing all templates. For each $Y^{(k)}$, some mechanism $g^{(k)}$ is needed to adaptively combine the representations of $\mathbf{X}$ to obtain $\mathbf{h}^{(k)}$, which is then fed into a classifier $f_{\mathrm{out}}^{(k)}$ to predict $\mathbf{Y}^{(k)}$.

The key component of this framework is the mechanism to produce $\mathbf{h}^{(k)}$, which we will design. The design assumes that $\mathbf{h}^{(k)}$ is a convex combination of the elements in $\mathcal{A}$, which allows us to consider $\mathbf{h}^{(k)}$ as the mean of an unknown distribution $p^{(k)}$ on $\mathcal{A}$, namely,

$$\boldsymbol{h}^{(k)} = \int_{\mathcal{A}} \mathbf{a} p^{(k)}(\mathbf{a}) \, \mathrm{d}\mathbf{a}. \tag{1}$$

In practice, the cardinality of $\mathcal{A}$ may be huge or infinite; therefore, it is important to design a mechanism that allows the users to inject prior knowledge to guide the production of $\mathbf{h}^{(k)}$. For example, in **TP**, $\mathcal{A}$ would be the set of all word embeddings, which could contain more than 10K elements. However, $\mathbf{h}^{(k)}$ should largely depend on the templates associated with the words (in the input sentence) having similar locations to the $k$-th word in the target sentence. If we could effectively inject this prior information, the inference task would be largely simplified.[2] One natural way to do so is to use a neural network module $f_{\mathrm{pref}}^{(k)}$ to propose a prototype of $p^{(k)}$, referred to as the preference distribution $u^{(k)}$, and let $p^{(k)}$ be close $u^{(k)}$. Specifically $u^{(k)}$ puts non-zero probability masses on templates $\mathbf{t}_1^{(k)}, \mathbf{t}_2^{(k)}, \ldots, \mathbf{t}_M^{(k)}$, and their probabilities are respectively $u_1^{(k)}, u_2^{(k)}, \ldots, u_M^{(k)}$ (which sum to 1). For the Translation Problem, $u^{(k)}$ is expected to have larger values for the words in a similar location of the $k$-th word of the target sentence. The preference distribution $u^{(k)}$ is considered as a good approximation of $p^{(k)}$, in the sense that the support of $p^{(k)}$ is contained in the set $\mathbf{T}^{(k)}$ of templates. Note that if $\mathbb{R}^d$ is the word embedding space for a large vocabulary, and if the size $M$ of the template set $\mathbf{T}^{(k)}$ is relatively small, then restricting the support of $p^{(k)}$ to be within $\mathbf{T}^{(k)}$ imposes a strong constraint on $p^{(k)}$. On the other hand, $u^{(k)}$ is not a sufficiently accurate approximation of $p^{(k)}$, in the sense that $u^{(k)}$ may assign probabilities to $\mathbf{T}^{(k)}$ somewhat differently. For example, in **TP**, the choice of $Y^{(k)}$

---

[1]We add the superscript $k$ to note that the inference of $Y^{(k)}$ does not necessarily share the set of templates.

[2]While we mainly use **Translation Problem (TP)** to motivate/justify our design and discussion, we will show the same idea also applies to the other two running examples in Sec 3.2.

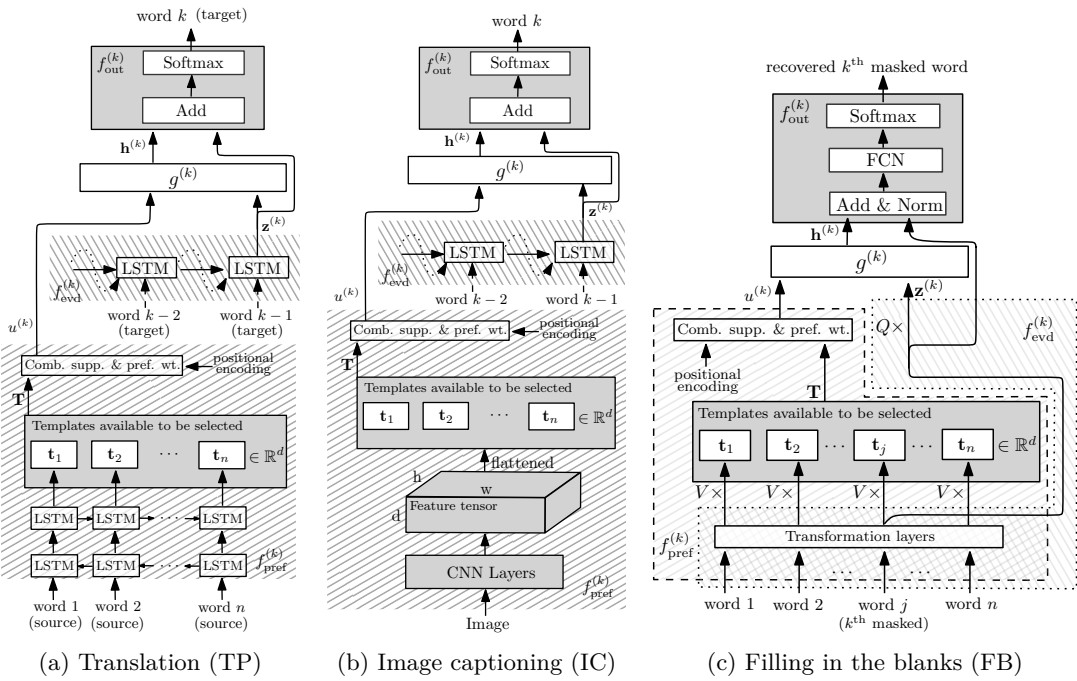

(a) Translation (TP)  (b) Image captioning (IC)  (c) Filling in the blanks (FB)

Figure 2: The model architectures of the three running examples. For the $f_{\text{evd}}^{(k)}$ in (a) and (b), the dashed links exist throughout the training and are replaced by the dotted ones in the generation stage.

depends on both $\mathbf{X}$ and the already generated words $Y^{(i<k)}$.[3] While $u^{(k)}$ provides a strong prior that $p^{(k)}$ should mainly focus on the words appearing in the source sentence, it is inherently tough for $u^{(k)}$ to capture the semantic evolution in $Y^{(i<k)}$. The difficulty shifts the mean $\boldsymbol{\mu}^{(k)}$ of $u^{(k)}$ from the mean $\mathbf{h}^{(k)}$ of $p^{(k)}$.

To alleviate the problem, we need another piece of information $\mathbf{z}^{(k)} \in \mathbb{R}^d$ that is generated by another network module $f_{\text{evd}}^{(k)}$ and provides information regarding the mean shift. (In **TP**, $\mathbf{z}^{(k)}$ depends on $Y^{(i<k)}$.) In particular, we assume that $\mathbf{z}^{(k)}$ is a noisy version of the shift, more precisely,

$$\mathbf{z}^{(k)} = \mathbf{h}^{(k)} - \boldsymbol{\mu}^{(k)} + \epsilon, \tag{2}$$

where $\epsilon \sim \mathcal{N}(\mathbf{0}, \sigma^2\mathbf{I})$ is a spherical Gaussian noise in $\mathbb{R}^d$ with covariance $\sigma^2\mathbf{I}$. We refer to $\mathbf{z}^{(k)}$ as the evidence.

We summarize the problem setup in Fig 1b. Then the design problem is *to construct a function, or a network block, g, which infers the unknown distribution $p^{(k)}$ and hence its mean $\mathbf{h}^{(k)}$ based on the evidence $\mathbf{z}^{(k)}$ and the preference distribution $u^{(k)}$.*

### 3.2 Practical examples

In this section, we first provide more details on how the setup applies to the transition problem (**TP**). Then we will show that the same setup also applies to the other two running examples.

**Translation Problem (TP).** For the translation problem, consider the model implementation plotted in Fig 2a that is similar to the one proposed in (Bahdanau et al., 2014). We will focus on the part of the model responsible for inferring the $k^{\text{th}}$ word of the target sentence. In this model, $\mathbf{h}^{(k)}$ corresponds to the constructed feature according to (1) that serves as an estimate of the context vector collecting the source sentence information. The estimated $\mathbf{h}^{(k)}$ is then fed into a classifier $f_{\text{out}}^{(k)}$ to predict the $k^{\text{th}}$ word. The preference distribution $u^{(k)}$ is generated by $f_{\text{pref}}^{(k)}$ which takes the source sentence words as inputs. In particular, the support of $u^{(k)}$ consists of the source sentence word embeddings $\mathbf{T}$ (called annotations

---

[3]We assume the sentence generation process is Markovian. More details are given in Sec 3.2.

in (Bahdanau et al., 2014)) which are pre-processed by two LSTM layers.[4] The preference weight for each template depends on some positional encoding functions, which, in principle, should assign higher weights to the templates appearing in the similar locations to the words we are inferring (that is, $\mathbf{h}^{(k)}$ is assumed to rely on the templates near $\mathbf{t}_k$ more heavily).

Note that the inferred $p^{(k)}$'s support must be a subset of $u^{(k)}$'s as it is reasonable to assume that the target sentence words only depend on those appearing in the source sentence. Besides, although the preference weights specified by the positional encoding functions could provide some *a priori* information for the templates' weights in $p^{(k)}$, they cannot be accurate as their inferences do not consider the previously generated words $Y^{(i<t)}$. This results in the mean $\boldsymbol{\mu}^{(k)}$ shifted from $\mathbf{h}^{(k)}$, which is estimated by $\mathbf{z}^{(k)} = f_{\text{evd}}^{(k)}$ that takes all the previously generated words $Y^{(i<t)}$ into account using another LSTM layer. Thus, $\mathbf{h}^{(k)}$ and $p^{(k)}$ should not be far from $\mathbf{z}^{(k)} + \boldsymbol{\mu}^{(k)}$ and $u^{(k)}$, respectively.

**Image Captioning (IC).** The caption generation model presented in Fig 2b has a similar architecture reported in (Xu et al., 2015). This model shares the designs of $f_{\text{evd}}^{(k)}$ and $f_{\text{out}}^{(k)}$ with the translation model while $f_{\text{pref}}^{(k)}$ instead extracts the templates from a raw image using a CNN network. In general, a word's position in the caption is independent of the location of the object it describes in the image. Therefore, in this model, all templates extracted by the CNN share the same preference weight.

As similar objects appear in an image would have similar features extracted by the CNN (for example, a zebra and a horse), allowing similar templates not in $\mathbf{T}$ to participate in $\mathbf{h}^{(k)}$'s estimation would possibly mix in information not contained in the raw image and harm the word inference accuracy. Therefore, we could improve the estimate of $\mathbf{h}^{(k)}$ by choosing $p^{(k)}$ similar to $u^{(k)}$ in the sense that $p^{(k)}$'s support cannot contain elements not in $u^{(k)}$'s.

Intuitively, as the generation process proceeds, the context $\mathbf{h}^{(k)}$ should be updated to provide relevant information in the image to facilitate the next word inference. Such change is governed by the caption's semantic evolution, which is captured by $\mathbf{z}^{(k)} = f_{\text{evd}}^{(k)}$ that predicts the shift of the mean $\boldsymbol{\mu}^{(k)}$ from $\mathbf{h}^{(k)}$. For this reason, $\boldsymbol{\mu}^{(k)} + \mathbf{z}^{(k)}$ serves as an estimate of $\mathbf{h}^{(k)}$ and should not be far away from it. Likewise, $u^{(k)}$ should be close to $p^{(k)}$.

**Filling in the Blanks Task (FB).** For the filling-in-the-blank tasks, let us consider a model architecture plotted in Fig 2c that is similar to the one used in BERT (Devlin et al., 2019). We focus on the inference of the $k^{\text{th}}$ masked word, which is assumed to be the $j^{th}$ word of the input sentence. In this model, $f_{\text{pref}}^{(k)}$ and $f_{\text{evd}}^{(k)}$ share the transformation layers (TL) that are commonly used in the natural language processing (NLP) tasks to map one sequence of vector representations to another of the same length.[5] Taking the output sequence, $f_{\text{pref}}^{(k)}$ applies a linear map $V$ to each of its elements to form $\mathbf{T}$ as the support of $u^{(k)}$ while the preference weights are specified by some positional encoding functions. At the same time, $\mathbf{z}^{(k)} = f_{\text{evd}}^{(k)}$ estimates $\mathbf{h}^{(k)}$'s shift from the mean $\boldsymbol{\mu}^{(k)}$ due to the variation of the local information. For the same reasons discussed in the previous two examples, we need $\boldsymbol{\mu}^{(k)} + \mathbf{z}^{(k)}$ close to $\mathbf{h}^{(k)}$ while $p^{(k)}$ is close to $u^{(k)}$.

Notably the formulation of the problem is based on the assumption that the network modules $f_{\text{evd}}^{(k)}$ and $f_{\text{pref}}^{(k)}$ are fixed and generate $\mathbf{z}^{(k)}$ and $u^{(k)}$ satisfying the above assumed properties. In reality, $f_{\text{evd}}^{(k)}$ and $f_{\text{pref}}^{(k)}$ are obtained via training. However, we argue that if $g$ is made to satisfy our design objective, we can at least *interpret* $f_{\text{evd}}^{(k)}$ and $f_{\text{pref}}^{(k)}$ obtained from training as serving to produce $\mathbf{z}^{(k)}$ and $u^{(k)}$ with our desired properties.

## 4 Formulation of an optimization problem

The discussion made in the previous section implies that the key optimization problem we are about to focus on should ensure

---

[4]In this model, given input $X$, all $u^{(k)}$'s share the same support $\mathbf{T}$. The superscripts of the templates are then omitted to show their independence from $k$. Similar comments apply to implementations of the other two running examples.

[5]Typical implementation of such layers include convolution layers, recurrent layers and self-attention layers.

1. $\mathbf{h}^{(k)}$ is not too far from $\boldsymbol{\mu}^{(k)} + \mathbf{z}^{(k)}$, where $\mathbf{h}^{(k)}$ is constructed by $p^{(k)}$ according to (1) and $\boldsymbol{\mu}^{(k)}$ is the mean of the preference distribution $u^{(k)}$.

2. $p^{(k)}$ is close to $u^{(k)}$ while $p^{(k)}$'s support is a subset of $u^{(k)}$'s.

These two desiderata prompt us to optimize:

$$\min_p \frac{\alpha}{2} \left\| (\boldsymbol{\mu} + \mathbf{z}) - \int_{\mathbb{R}^d} \mathbf{a} p(\mathbf{a}) \, \mathrm{d}\mathbf{a} \right\|^2 + \mathcal{K}(p, u) \tag{3}$$

where $\alpha > 0$ is responsible for the relative strength of the two terms (and can be interpreted as the reliability of $\boldsymbol{\mu} + \mathbf{z}$), $\mathcal{K}(p, u)$ denotes the KL divergence from $p$ to $u$.[6] By definition, $\mathcal{K}(p, u)$ has a finite value if and only if $p$ has zero values outside the support of $u$. Thus, both requirements in the second desideratum are satisfied by using the KL divergence as a measure for the closeness of $p$ and $u$. Let $\tilde{p}$ be the minimizer of (3). The estimate of $\mathbf{h}$ is

$$\hat{\mathbf{h}} = \int_{\mathbb{R}^d} \mathbf{a} \tilde{p}(\mathbf{a}) \, \mathrm{d}\mathbf{a}. \tag{4}$$

Naturally, this optimization problem can be derived from three different, though, related perspectives. We present the one we believe is not widely known below and include the maximum likelihood and Bayesian perspectives in Appx B.

**A Maximum Entropy on the Mean Perspective.** Consider a problem that seeks a distribution $p$ such that the expectation $\int_{\mathbb{R}^d} \mathbf{a} p(\mathbf{a}) \, \mathrm{d}\mathbf{a}$ is not far from $\boldsymbol{\mu} + \mathbf{z}$. In particular, we require

$$\left\| (\boldsymbol{\mu} + \mathbf{z}) - \int_{\mathbb{R}^d} \mathbf{a} p(\mathbf{a}) \, \mathrm{d}\mathbf{a} \right\|^2 \leq \frac{1}{2\alpha}. \tag{5}$$

Note that, given $\mathbf{z}$, there are infinitely many $p$'s that satisfy the constraints, which makes it difficult to pick a "best" $p$ for later use. A technique known in information theory as the maximum entropy on the mean (MEM) (Rioux et al., 2020; Gamboa, 1989) solves this problem by picking the best guess of the ground truth $p^*$ that simultaneously satisfies (5) and minimizes the KL divergence to the distribution $u$. That is,

$$\tilde{p} = \operatorname*{argmin}_p \mathcal{K}(p, u) \text{ s.t. } \left\| (\boldsymbol{\mu} + \mathbf{z}) - \int_{\mathbb{R}^d} \mathbf{a} p(\mathbf{a}) \, \mathrm{d}\mathbf{a} \right\|^2 \leq \frac{1}{2\alpha},$$

which is also the minimizer of (3) according to Equation (18) of (Rioux et al., 2020) and Corollary 4.9 of (Borwein & Lewis, 1992).

## 5 A motivating example to find the optimal solution

We first show how to solve (3) when the preference distribution $u$ is spherical Gaussian. That is, $u \sim \mathcal{N}(\boldsymbol{\mu}, I_d)$.

Let $\mathbf{b} = \boldsymbol{\mu} + \mathbf{z}$ serve as an unreliable observation of $\mathbf{h}_p$. Rioux et al. (Rioux et al., 2020) prove, via Fenchel duality (Rockafellar, 1970, p.104) that the minimizer $p^*$ of (3) takes the form

$$p^*(\mathbf{a}) = \frac{u(\mathbf{a}) \exp\langle \mathbf{a}, \boldsymbol{\lambda}^* \rangle}{\int u(\mathbf{a}') \exp\langle \mathbf{a}', \boldsymbol{\lambda}^* \rangle \, \mathrm{d}\mathbf{a}'}, \tag{6}$$

where

$$\boldsymbol{\lambda}^* = \operatorname*{argmax}_{\boldsymbol{\lambda} \in \mathbb{R}^d} \langle \mathbf{b}, \boldsymbol{\lambda} \rangle - \frac{1}{2\alpha} \|\boldsymbol{\lambda}\|^2 - \log \int u(\mathbf{a}) \exp\langle \mathbf{a}, \boldsymbol{\lambda} \rangle \, \mathrm{d}\mathbf{a}. \tag{7}$$

Note that $\int u(\mathbf{a}) \exp\langle \mathbf{a}, \boldsymbol{\lambda} \rangle \, \mathrm{d}\mathbf{a} = \exp(\langle \boldsymbol{\mu}, \boldsymbol{\lambda} \rangle + \frac{1}{2} \|\boldsymbol{\lambda}\|^2)$ as it is the moment generating function (MGF) of $u \sim \mathcal{N}(\boldsymbol{\mu}, I_d)$. Substituting the expression into (7) followed by setting the derivative with respect to $\boldsymbol{\lambda}$ to

---

[6]As we will focus on a single step of sequence predictions, we simplify our notations by omitting superscript $(k)$ in the rest of our discussions.

zero yields $\boldsymbol{\lambda}^* = \frac{\alpha}{\alpha+1}(\boldsymbol{b} - \boldsymbol{\mu})$. By (6), $p^*(\mathbf{a}) \propto \exp(-\frac{1}{2}\|\mathbf{a} - \boldsymbol{\mu}\|^2 + \langle\mathbf{a}, \boldsymbol{\lambda}^*\rangle) \propto \exp(-\frac{1}{2}\|\mathbf{a} - (\boldsymbol{\mu} + \boldsymbol{\lambda}^*)\|^2)$. Substituting $\boldsymbol{\lambda}^* = \frac{\alpha}{\alpha+1}(\boldsymbol{b} - \boldsymbol{\mu})$ into it implies that $p^*$ follows a Gaussian distribution $\mathcal{N}(\frac{1}{1+\alpha}\boldsymbol{\mu} + \frac{\alpha}{1+\alpha}\boldsymbol{b}, I_d)$. Thus, our estimate of $\mathbf{h}_p$ is $\frac{1}{1+\alpha}\boldsymbol{\mu} + \frac{\alpha}{1+\alpha}\mathbf{b}$.

The value $\alpha$ in (3) can also be considered as a measure of the reliability of the noisy observation $\boldsymbol{b}$, where a smaller $\alpha$ implies a less reliable $\boldsymbol{b}$. Then, the estimate of $\boldsymbol{h}_p$ should be less affected by $\boldsymbol{b}$ as $\alpha$ approaches zero, which is well captured by our derived expression $\frac{1}{1+\alpha}\boldsymbol{\mu} + \frac{\alpha}{1+\alpha}\mathbf{b}$. We will also see this relationship in a more general setting in our subsequent discussions. While a more complicated analysis is involved, the underlying principles are essentially the same.

In Sec 6, we focus on a similar optimization problem that estimates $\mathbf{h}_p$ assuming that $u$ is instead a discrete distribution. By solving the optimization problem, we derive a closed-form approximation for the estimate of $\mathbf{h}_p$, via Fenchel duality. The approximation then gives a generalized attention layer structure as shown in Fig 1. A special case of it is equivalent to the familiar dot-product attention (Luong et al., 2015) that is also adopted in transformers (Vaswani et al., 2017). Moreover, we will show that T5 transformer (Raffel et al., 2020) implicitly adopts our generalized attention expression.

## 6 Optimal solution

Rioux et al. proved that the optimization problem stated in (3) has the following Fenchel dual (see Theorem 2 of (Rioux et al., 2020)):

**Theorem 1.** *The dual of (3) is given by*

$$\max_{\boldsymbol{\lambda} \in \mathbb{R}^d} \left\{ \langle\boldsymbol{\lambda}, \boldsymbol{\mu} + \mathbf{z}\rangle - \frac{1}{2\alpha}\|\boldsymbol{\lambda}\|^2 - \log M(\boldsymbol{\lambda}) \right\}, \tag{8}$$

*where*

$$M(\boldsymbol{\lambda}) = \int_{\mathbb{R}^d} u(\mathbf{a}) \exp\langle\mathbf{a}, \boldsymbol{\lambda}\rangle \, d\mathbf{a}. \tag{9}$$

*Given a maximizer $\boldsymbol{\lambda}^*$ of (8), one can recover the minimizer $\tilde{p}$ of (3) via*

$$\tilde{p}(\mathbf{a}) = \frac{u(\mathbf{a}) \exp\langle\mathbf{a}, \boldsymbol{\lambda}^*\rangle}{\int_{\mathbb{R}^d} u(\mathbf{a}') \exp\langle\mathbf{a}', \boldsymbol{\lambda}^*\rangle \, d\mathbf{a}'}. \tag{10}$$

By Theorem 1, the estimated $\mathbf{h}$ defined in (4) can be re-written as

$$\hat{\mathbf{h}} = \int_{\mathbb{R}^d} \mathbf{a}\tilde{p}(\mathbf{a}) \, d\mathbf{a} = \int_{\mathbb{R}^d} \mathbf{a} \frac{u(\mathbf{a}) \exp\langle\mathbf{a}, \boldsymbol{\lambda}^*\rangle}{\int_{\mathbb{R}^d} u(\mathbf{a}') \exp\langle\mathbf{a}', \boldsymbol{\lambda}^*\rangle \, d\mathbf{a}'} \, d\mathbf{a}, \tag{11}$$

where $\boldsymbol{\lambda}^*$ is a maximizer of (8).

In general, $\boldsymbol{\lambda}^*$ does not have a closed-form expression in terms of $\alpha$, $u$ and $\mathbf{z}$, and a standard paradigm is to search for it using gradient ascent-based methods. In this paper, we will not search for $\boldsymbol{\lambda}^*$ in this way; instead, we will derive a closed-form expression to approximate it. Remarkably, this takes the form of the generalized attention presented in Fig 1.

Note that $M(\boldsymbol{\lambda})$ in (9) equals $\mathbb{E}_u[\exp\langle W, \boldsymbol{\lambda}\rangle]$, the expectation of the random variable $\exp\langle W, \boldsymbol{\lambda}\rangle$ where $W$ has the probability distribution $u$. The expectation is just the moment generating function (MGF) of $W$, and the value $\log M(\boldsymbol{\lambda})$ is called the cumulant of $W$ (McCullagh, 1987, p.26), which has an expansion (McCullagh, 1987, (2.4))

$$\log M(\boldsymbol{\lambda}) = \langle\boldsymbol{\mu}, \boldsymbol{\lambda}\rangle + \frac{1}{2}\langle\boldsymbol{\lambda}, \Sigma\boldsymbol{\lambda}\rangle + o(\|\boldsymbol{\lambda}\|^2), \tag{12}$$

with $\boldsymbol{\mu} = \int \mathbf{a}u(\mathbf{a}) \, d\mathbf{a}$ and $\Sigma = \int (\mathbf{a} - \boldsymbol{\mu})(\mathbf{a} - \boldsymbol{\mu})^T u(\mathbf{a}) d\mathbf{a}$ respectively denote the expectation and the variance-covariance matrix of $W$. Note that the expansion implicitly assumes that random variable $W$ following distribution $u$ has bounded moments. (Derivation of (42) is given in A.)

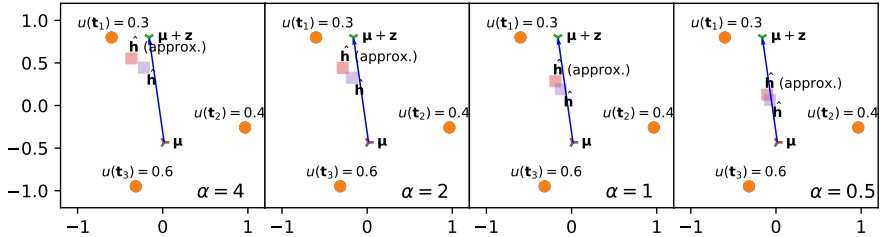

Figure 3: The approximation of $\hat{\mathbf{h}}$ for different choices of $\alpha$. The dots in orange compose the support of discrete $u$ with the preference weights labelled above. The dark blue arrow starting from the mean $\mu$ of $u$ denotes the evidence $\mathbf{z}$. The red square marks the $\hat{\mathbf{h}}$ constructed by (11) with the $\boldsymbol{\lambda}^*$ maximizing (8), while the purple one marks the $\hat{\mathbf{h}}$ approximated by (17). As we can observe, (17) gives a precise approximation of $\hat{\mathbf{h}}$ when $\alpha$ is sufficiently small.

Now we assume that $\alpha$ is small and we argue that this assumption is justified in practice. For instance, in the translation task, all words in the dictionary can serve as candidate templates, which could be more than 10,000, but $u$ reduces this size to the length of the source sentence (usually less than tens of words). The inference of $p$ should strongly anchor around this prior information; consequently the information provided by $\mathbf{z}$ should weigh less. On the other hand, $\mathbf{z}$ can hardly provide an accurate estimate of the mean shift, since the generation of $\mathbf{z}$ is often ignorant of the templates selected by $u$ (for example, in the example translation and image captioning models) or generated by a low-capacity module (as in the example filling-in-the-blank model). For these reasons, one should de-emphasize the constraint imposed by $\mathbf{z}$ and thus choose a small $\alpha$.

When $\alpha$ is picked to be small enough (see (8)), the optimization of $\boldsymbol{\lambda}$ gets a large penalty on its L2 norm and thus, $\|\boldsymbol{\lambda}^*\|$ is close to zero. Then, by (42), we have

$$\log M(\boldsymbol{\lambda}^*) \approx \langle \mu, \boldsymbol{\lambda}^* \rangle + \frac{1}{2} \langle \boldsymbol{\lambda}^*, \Sigma \boldsymbol{\lambda}^* \rangle. \tag{13}$$

Note that the approximation becomes exact for any $\alpha > 0$ if $u$ is Gaussian, which is the case of the motSec ivating example in Sec 5. Substituting (13) into (8) followed by setting the derivative with respect to $\boldsymbol{\lambda}$ to zero yields

$$\boldsymbol{\lambda}^* = \alpha(I_d + \alpha\Sigma)^{-1}\mathbf{z}, \tag{14}$$

where $I_d$ denotes the $d \times d$ identity matrix.[7] As $\alpha$ is assumed close to zero, (14) is further reduced to

$$\boldsymbol{\lambda}^* = \alpha\mathbf{z}. \tag{15}$$

Plugging the expression into (11) gives the result stated as follows:

**Theorem 2.** *Given $u$ with bounded moments, for a small enough $\alpha > 0$, the estimated $\mathbf{h}$ defined in (4) can be approximated by*

$$\hat{\mathbf{h}} = \int_{\mathbb{R}^d} \mathbf{a} \, \frac{u(\mathbf{a})\exp(\alpha\langle\mathbf{a},\mathbf{z}\rangle)}{\int_{\mathbb{R}^d} u(\mathbf{a}')\exp(\alpha\langle\mathbf{a}',\mathbf{z}\rangle)\,\mathrm{d}\mathbf{a}'} \, \mathrm{d}\mathbf{a}. \tag{16}$$

For the case that $u$ is a discrete distribution with support $\{\mathbf{t}_1, \mathbf{t}_2, \ldots, \mathbf{t}_n\}$ and the preference probability $\{u_1, u_2, \ldots, u_n\}$, (16) becomes simply

$$\hat{\mathbf{h}} = \sum_{i=1}^{n} \mathbf{t}_i \, \frac{u_i \exp\left(\alpha\langle\mathbf{t}_i, \mathbf{z}\rangle\right)}{\sum_{j=1}^{n} u_j \exp\left(\alpha\langle\mathbf{t}_j, \mathbf{z}\rangle\right)}. \tag{17}$$

In Fig 3, we set $d = 2$ and visualize the approximation of $\mathbf{h}$ for various selections of $\alpha$. We observe that, as $\alpha$ decreases, (17) outputs a better approximation of $\hat{\mathbf{h}}$. Besides, as a decreasing $\alpha$ implies a less reliable $\boldsymbol{\mu}+\boldsymbol{z}$, $\boldsymbol{h}$

---

[7]When $\Sigma = I_d$, (14) becomes $\boldsymbol{\lambda}^* = \alpha(I_d + \alpha I_d)^{-1}\mathbf{z} = \frac{\alpha}{1+\alpha}\mathbf{z}$. By (2), $\mathbf{b} = \mathbf{h} + \epsilon = \mathbf{z} + \boldsymbol{\mu}$. Thus, $\boldsymbol{\lambda}^* = \frac{\alpha}{1+\alpha}(\boldsymbol{b} - \boldsymbol{\mu})$ recovers the expression of $\boldsymbol{\lambda}^*$ in the motivating example.

is less affected by $\boldsymbol{\mu}+\boldsymbol{z}$ and gets close to $\boldsymbol{\mu}$. Note that our results do not suggest that $\alpha$ should be arbitrarily close to zero for a perfect approximation (which leaves $\mathbf{z}$ useless). Fig 3 shows a good approximation is achieved when $\alpha = 0.5, 1$. And for these two choices, $\hat{\boldsymbol{h}}$ still significantly deviates from $\boldsymbol{\mu}$ (corresponding to the case when $\alpha = 0$ and $\mathbf{z}$ is useless). Thus, $\mathbf{z}$ still largely affects the final estimation results.

In Sec 8, we will show that a good approximation can be made in practice by comparing the accurate solution with its approximated counterpart used in the pretrained BERT model (Devlin et al., 2019) and T5 model (Raffel et al., 2020).

## 7 Discussion

In Section 6, we derived an alternative expression of $\hat{\mathbf{h}}$ defined in (4) by solving the Fenchel dual of the optimization problem (3). Although the expression is not in closed form, as we are only interested in the case when $\alpha$ is small, a closed-form approximation of $\hat{\mathbf{h}}$ is derived in Theorem 2 and reduced to the form stated in (17) when considering a discrete distribution $u$.

As we pointed out, the block $g$ in Fig 2a, Fig 2b and Fig 2c is expected to find the inferred $\tilde{p}$ minimizing (3) followed by plugging it into (4) to construct $\hat{\mathbf{h}}$. Thus, one can complete the architecture designs of the three running examples by replacing $g$ with a network layer implementing (17), namely, the structure in Fig 1c.

**The relationship between the optimal solution and attention models.** Remarkably, the expression stated in (17) gives a generalized attention block. In particular, based on our framework, researchers can customize the implementations of $f_{\mathrm{evd}}^{(k)}$ and $f_{\mathrm{pref}}^{(k)}$ to generate $\mathbf{z}$ and $u$ and feed them into (17) to get an attention-like network architecture.[8]

For instance, by setting $u_i = \frac{1}{n}$ for all $i$, the expression is equivalent to the well known dot-product attention (Luong et al., 2015), which is also applied in the transformer network (Vaswani et al., 2017). The equivalence of the expression of $\hat{\mathbf{h}}$ and the dot-product attention layer tells us: (a) *by applying a dot-product attention layer in a model, we essentially ask the model to perform an optimization task defined in (3) and construct the output according to (4).* (b) *the derivation of h depends on two relatively independent pieces of information: a preference distribution given the global information and an estimate of the output's deviation from the preference distribution's mean according to some local information. This suggests that the design of attention-based model can be decomposed into two parts that respectively estimate these two values.*

**The model consisting of a stack of attention layers.** Although our discussion focuses on the case that contains a single attention layer, any attention layer $\mathcal{L}$ in an attention stack fits our framework (see Fig 1). In particular, all the attention layers closer to the input $X$ than $\mathcal{L}$ can be grouped into the functions $f_{\mathrm{pref}}$ or $f_{\mathrm{evd}}$. For those layers that take the current layer's output as input, we can group them into $f_{\mathrm{out}}$, where $\mathbf{c}$ may contain the outputs of other attention layers working in parallel.

**T5 transformer implicitly adopts the generalized attention structure.** Recent studies in NLP have shown that T5 transformer (Raffel et al., 2020) can achieve state-of-the-art performance for many NLP benchmarks, including text summarization, classification, question answering, etc. While their transformer implementations are quite similar to the original transformer architecture (Vaswani et al., 2017; Devlin et al., 2019), they adopt trainable relative position embeddings to replace the sinusoidal position signals.[9] The modification provides the model with extra flexibility to encode the positional information with little computational cost.

We will see that in comparison to the original transformer implementation, T5 transformer can be seen as a natural realization of the generalized attention in (17), where the preference weights $u$ unifies the concepts of word masks and T5's positional encoding functions. As a result, the usefulness and the validity of our framework are well-supported by the state-of-the-art performance of T5 in many NLP tasks (Raffel et al., 2020).

---

[8]Potential selectionss of $f_{\mathrm{evd}}^{(k)}$ and $f_{\mathrm{pref}}^{(k)}$ includes constant functions, fixed formulas and neural networks.

[9]They also simplified the layer normalization (Lei Ba et al., 2016) for faster training and inference speed.

Consider the running example: filing in the blanks, with the preference distribution

$$u(\mathbf{t}_i) = \begin{cases} 0 & \text{if the } i^{\text{th}} \text{ word is masked} \\ \exp(b_{j-i})/Z & \text{otherwise,} \end{cases} \tag{18}$$

where $Z$ is a normalizing constant and $b_{j-i}$ is a trainable scalar that only depends on the relative position of word $i$ and word $j$ (which is the $k^{\text{th}}$ masked word that we are inferring). Substituting such $u$ into (17) with $\alpha = 1$ yields

$$\hat{\mathbf{h}} = \sum_{i=1}^{n} \mathbf{t}_i \frac{\exp\left(\langle \mathbf{t}_i, \mathbf{z}\rangle + b_{j-i} + \mathbf{1}_{\text{masked}}(i)\right)}{\sum_{l=1}^{n} \exp\left(\langle \mathbf{t}_l, \mathbf{z}\rangle + b_{j-l} + \mathbf{1}_{\text{masked}}(l)\right)}, \tag{19}$$

where $\mathbf{1}_{\text{masked}}(i)$ is an indicator function that equals $-\infty$ if word $i$ is masked and zero otherwise. The expression in (19) has the same structure as that adopted in T5 transformer, where the indicator function serves as the mask function to prevent the model from assigning weights to the masked words. In this way, the concepts of word masks and the positional encoding functions are unified by $u$ in (18). Conversely, T5 transformer is a realization of the generalized attention with the preference weights $u$ specified in (18).

**Generalized attention structures suggested by the optimal solution.** While T5 transformer has implicitly adopted the generalized attention, (17) hints further generalizations could be made. For instance, in T5 transformer, the function outputting template's preference weights only considers the word masks and the word's relative positions. This function could be generalized to also consider the input sentence contexts, and the output weights encode the importance of each word before giving the local information stored in $\mathbf{z}$. The same idea could be applied to the image captioning example to replace the uniform preference weights. By adding a neural network taking the input image to generate non-uniform preference weights, we devise a mechanism to estimate the importance of each part of the image before the caption generation. In this way, the newly added network collects global information from the image to propose a preference distribution, which could be updated locally based on current generation stage encoded in $\mathbf{z}$.

Besides, although we mainly focus on the case when $u$ is discrete, we want to emphasize that the analysis performed in Section 6 also covers continuous $u$. This hints that a continuous attention mechanism could also be implemented, which might prove to be useful in some applications.

Moreover, our theoretical work enables the design of more general attention structures; for instance, KL-divergence in the optimization problem (3) requires estimated distribution to share support with preference distribution, which may not be desired in many tasks. (e.g. translation, where the target should be unaffected if we replace some words in the source sentence with synonyms.) Using our theory, in Sec 9, we show that this can be achieved by replacing KL divergence with an optimal transport (OT)-based measure that handles word similarities in their embedding space.

## 8 Empirical evidence

To show the proposed optimization problem (3) indeed provides a principle justifying the design of attention modules, we show that the maximizer $\boldsymbol{\lambda}^*$ of its dual problem (8) nearly coincides with its approximated counterpart used in the pretrained BERT model (Devlin et al., 2019) and T5-small (Raffel et al., 2020). Verification on other popular attention-based models yielded similar results.

Let $\mathbf{x}_i \in \mathbb{R}^d$ for $i \in 1, 2 \ldots, n$, $\mathbf{y}_j \in \mathbb{R}^d$ for $i \in 1, 2 \ldots, m$ and $K, Q, V \in \mathbb{R}^{d' \times d}$. The $k^{\text{th}}$ output of BERT attention is

$$\sum_{i=1}^{n} V\mathbf{x}_i \; \frac{\exp\left(\langle K\mathbf{x}_i, Q\mathbf{x}_k\rangle/\sqrt{d'}\right)}{\sum_{j=1}^{n} \exp\left(\langle K\mathbf{x}_j, Q\mathbf{x}_k\rangle/\sqrt{d'}\right)} \tag{20}$$

and the one for T5 is

$$\sum_{i=1}^{n} V\mathbf{x}_i \; \frac{u_i \exp\left(\langle K\mathbf{x}_i, Q\mathbf{y}_k\rangle/\sqrt{d'}\right)}{\sum_{j=1}^{n} \exp\left(\langle K\mathbf{x}_j, Q\mathbf{y}_k\rangle/\sqrt{d'}\right)}. \tag{21}$$

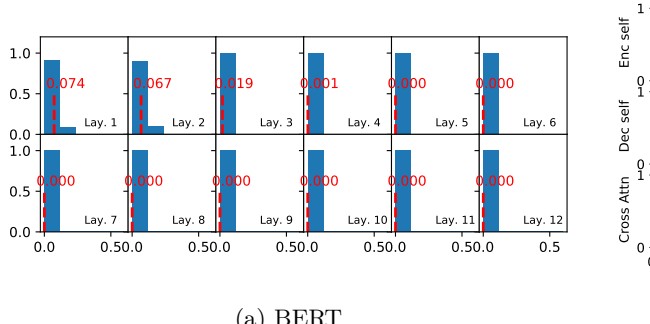 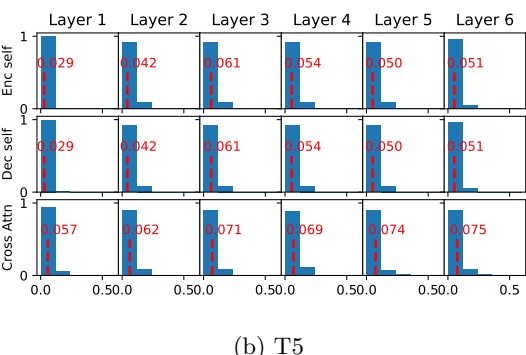

(a) BERT  (b) T5

Figure 4: The distribution of relative deviations $\frac{\|\boldsymbol{\lambda}^* - \alpha\mathbf{z}\|}{\|\boldsymbol{\lambda}^*\|}$ for the attention in BERT and T5. The red vertical lines mark the average of the errors.

T5 has three types of attention, self-attentions in the encoder and the decoder and the cross-attention connecting them. For the two self-attentions, $\mathbf{x}_i = \mathbf{y}_i$ and $m = n$.

For BERT, setting $\alpha = 1$, $\mathbf{t}_i = \frac{\mathbf{x}_i}{\sqrt{d'}}$, $\mathbf{z} = K^\top Q\mathbf{x}_k$, $V' = V\sqrt{d'}$ and $u_i \propto 1$ yields $V' \sum_{i=1}^{n} \mathbf{t}_i \frac{u_i \exp\langle\mathbf{t}_i, \mathbf{z}\rangle}{\sum_{j=1}^{n} u_j \exp\langle\mathbf{t}_j, \mathbf{z}\rangle}$, where the summation part is the one derived in (17).[10] Likewise, for T5, we use the same setting as the BERT's except that $u_i$ is computed based on its positional encoding and $\mathbf{z} = K^\top Q\mathbf{y}_k$ for the cross-attention.

We find $\boldsymbol{\lambda}^*$ by plugging $\alpha$, $u_i$'s, $\mathbf{t}_i$'s and $\mathbf{z}$ into (8) followed by performing gradient ascent. We then calculate the relative deviation $\frac{\|\boldsymbol{\lambda}^* - \alpha\mathbf{z}\|}{\|\boldsymbol{\lambda}^*\|}$ of its approximated counterpart $\alpha\mathbf{z}$ and report its distribution in Fig 4 for each attention layer by taking the average over the attention heads. We report the distributions for each head in C. As Fig 4 indicates, $\boldsymbol{\lambda}^*$ almost coincide with its approximated counterpart $\alpha\mathbf{z}$ inferred by BERT and T5, which corroborates that problem (3) gives a principle justifying the design of attention.

## 9 An optimal transport-based attention

In Sec 7, we mentioned that our theoretical work enables the design of more general attention structures. Let $\mathbb{R}_+$ denote the set of non-negative real numbers. In this section, we provide an example by replacing the KL-divergence in (3) with an entropy-regularized OT-based measure (Cuturi, 2013):

$$\mathcal{W}_\gamma(p, u; M) = \min_{X \in U(p,u)} \langle M, X \rangle - \gamma H(X), \tag{22}$$

where $\gamma > 0$, $H(X) = \sum_{i,j=1}^{N} -X_{ij} \log X_{ij}$ is the entropy of $X$, $U(p, u) = \{X \in \mathbb{R}_+^{|\mathcal{A}| \times |\mathcal{A}|}; X\mathbf{1} = p, X^T\mathbf{1} = u\}$ and $M \in \mathbb{R}_+^{|\mathcal{A}| \times |\mathcal{A}|}$ is a cost matrix that measures the similarity between each pair of the templates in $\mathcal{A}$.[11] The entropy regularization makes the minimizer $X^*$ in (22) change smoothly in terms of $p, u$ and $M$, which stabilizes and speeds up evaluation of $\mathcal{W}$ (Cuturi, 2013). When $\gamma \to 0$, $M(\mathbf{t}, \mathbf{t}') = d_\mathcal{A}(\mathbf{t}, \mathbf{t}')^\rho$, $\mathcal{W}_\gamma^{1/\rho}$ is reduced to the Wasserstein $\rho$-distance. We note that, due to the entropy term, for fixed $u$ and $M$, the true preference distribution $\tilde{u}$ that minimizes $\mathcal{W}_\gamma(\tilde{u}, u; M)$ is slightly deviated from $u$ and will approach to $u$ if $\gamma \to 0$. (see Appx D for details.) Let $\tilde{\boldsymbol{\mu}}$ denote the expectation of $\tilde{u}$. Then we can rewrite (3) as

$$\min_p \frac{\alpha}{2} \left\| (\tilde{\boldsymbol{\mu}} + \mathbf{z}) - \int_{\mathbb{R}^d} \mathbf{a} p(\mathbf{a}) \, d\mathbf{a} \right\|^2 + \mathcal{W}_\gamma(p, u; M). \tag{23}$$

---

[10]Templates $\mathbf{t}_i$'s absorb the scaling factor $d'^{-\frac{1}{2}}$ so that their norms remain bounded as $d'$ increases. Thus, $u$ has bounded moments, and Theorem 2 applies. Note that it is a common practice to scale outputs before performing theoretical analysis. (e.g. see the work of Arora et al. (2019).)

[11]A smaller $M_{ij}$ implies a larger similarity between $\mathbf{t}_i$ and $\mathbf{t}_j$. While many OT-related problems define $M$ by embedding templates into a metric space $(\mathcal{A}, d_\mathcal{A})$ with $M(\mathbf{t}, \mathbf{t}') = d_\mathcal{A}(\mathbf{t}, \mathbf{t}')^\rho$, $\rho \geq 1$, our discussion makes no assumption on $M$ other than it is non-negative and symmetric, and $M(\mathbf{t}, \mathbf{t}) < M(\mathbf{t}', \mathbf{t})$ for all $\mathbf{t}' \neq \mathbf{t}$.

Following a similar procedure presented in Sec 6 (the derivation is given in Appx D), we can derive and solve its Fenchel dual problem and show that when both $\alpha$ and $\frac{\alpha}{\gamma}$ are small, the minimizer $p^*$ takes the form

$$p^*(\mathbf{t}) = \sum_{i=1}^{n} u_i \, \exp\big(\left(\alpha \mathbf{t}^T \mathbf{z} - M(\mathbf{t}, \mathbf{t}_i)\right)/\gamma\,\big)\big/Z_i \tag{24}$$

with $Z_i = \sum_{\mathbf{t}' \in \mathcal{A}} \exp\left(\left(\alpha(\mathbf{t}')^T z - M(\mathbf{t}', \mathbf{t}_i)\right)/\gamma\right)$. Substituting (24) into (4), we get the OT-based attention.

**The OT-based attention considers all templates in $\mathcal{A}$.** In comparison to the generalized attention derived in Sec 6, the OT-based one assigns non-zero weights to the all templates in $\mathcal{A}$. To see how it works, consider an extreme case that the templates are partitioned into several groups. If two templates $\mathbf{t}, \mathbf{t}'$ belong to the same group, $M(\mathbf{t}, \mathbf{t}') = 0$; otherwise, $M(\mathbf{t}, \mathbf{t}) = \infty$. Moreover, templates within the same groups are very similar in a sense that their inner products with $\mathbf{z}$ are approximately equal. Suppose $\mathbf{t}_i$ belongs to a group $\mathcal{G}$ and other templates $\mathbf{t}_{j \neq i}$ do not, then for all $\mathbf{t} \in \mathcal{G}$, we have $p^*(\mathbf{t}) = u_i/|\mathcal{G}|$. That is, all templates of $\mathcal{G}$ share the weight of $\mathbf{t}_i$ and thus be potentially trained even if most of them do not appear in the input.

In general, if a template $\mathbf{t}$ is similar to some $\mathbf{t}_i \in \mathbf{T}$ (i.e., $M(\mathbf{t}, \mathbf{t}_i)$ is small), it will share $\mathbf{t}_i$'s weight although it does not appear in $\mathbf{T}$. In contrast, for a regular attention, only templates in $\mathbf{T}$ can be assigned non-zero weights. The peculiar property of the OT-based attention is desired in some practical tasks. For example, in an NLP problem, synonyms intuitively have similar templates. Then if a word appears in the input sentence and is trained, its synonyms should be trained in a similar way and thus been assigned a similar weight (because replacing a word with its synonym does not alter the input in a semantic sense).

## 10 Conclusion

This paper presented a new perspective to understand the attention mechanism by showing that it can be viewed as a solver of a family of inference tasks. These tasks involve improving the noisy estimate of a distribution $p$'s mean by a preference distribution that encodes some beliefs of $p$'s value. We have used three running examples with the typical model architectures to show that such tasks naturally exist in neural network design. We then abstracted a convex optimization problem from these tasks and derived a closed-form approximation of the optimal solution by solving the problem's Fenchel dual. We find that the closed-form approximation can be seen as a generalized attention layer and one of its special cases is equivalent to the dot-product attention adopted in transformers. We further performed an analysis on the general form and showed that T5 transformer implicitly adopts the generalized attention structure with attention weights unifying the concepts of the word masks and the positional encoding functions. We empirically show that our framework can well-explain the attention inference in the pretrained BERT and T5 models. To demonstrate how our work enables the design of more general attention structure, we replace the KL divergence with an OT-based measure and derive an OT-based attention structure, which frees the designer from the $p^{(k)}$ support constraints alluded to in the examples.

This paper presents a principled justification for the design of attention modules in neural networks. Specifically, there is a general assumption that because attention in humans narrows the search space, a similar phenomenon is at play in transformers. In this paper, we have shown that the mechanism corresponds to proposing a preference distribution over the templates, followed by adjusting it using a noisy mean shift estimation. The generalized attention structure presented potentially opens a door to a wide design space. For example, the preference weights need not be derived from the positional encoding functions; they could integrate a variety of information provided by other components of the network. Additionally, this research successfully demonstrates a novel approach to analyze the functioning of a neural network component, namely, via isolating the component from the complex network structure and asking: is there a "local problem" that is solved by the design of this component?

**Broader impact statement.** This paper presents a new perspective to understand attention and derived a generalized attention structure. Our work is foundational, which we believe does not have direct negative societal impacts. Due to the very wide range of applications of attention, such as self-driving (Kim & Canny, 2017), healthcare (Ma et al., 2017) and protein interaction prediction (Tsubaki et al., 2018), we expect our works can facilitate the algorithm developments in these areas, which may have unexpected impacts.

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

## A    Derivation of (42) for preference distributions of bounded moments

Assume a preference distribution $u$ has bounded moments. Then its moment generating function

$$M(\boldsymbol{\lambda}) = \int_{\mathbb{R}^d} \langle \mathbf{a}, \boldsymbol{\lambda} \rangle u(\mathbf{a}) \mathrm{d}\mathbf{a} = 1 + \langle M'(0), \boldsymbol{\lambda} \rangle + \frac{1}{2} \langle \boldsymbol{\lambda}, M''(0) \boldsymbol{\lambda} \rangle + o(\|\boldsymbol{\lambda}\|^2), \tag{25}$$

where

$$M'(0) = \int \mathbf{a} u(\mathbf{a}) \mathrm{d}\mathbf{a} = \boldsymbol{\mu}, \tag{26}$$

$$M''(0) = \int \mathbf{a}\mathbf{a}^\top u(\mathbf{a}) \mathrm{d}\mathbf{a}. \tag{27}$$

Notice that

$$\log(1 + x) = t - \frac{t^2}{2} + \frac{t^3}{3} - \frac{t^4}{4} + \cdots = t - \frac{t^2}{2} + o(t^2). \tag{28}$$

Thus,

$$\begin{aligned}
\log(M(\boldsymbol{\lambda})) &= \left( \langle M'(0), \boldsymbol{\lambda} \rangle + \frac{1}{2} \langle \boldsymbol{\lambda}, M''(0) \boldsymbol{\lambda} \rangle + o(\|\boldsymbol{\lambda}\|^2) \right) \\
&\quad - \frac{1}{2} \left( \langle M'(0), \boldsymbol{\lambda} \rangle + \frac{1}{2} \langle \boldsymbol{\lambda}, M''(0) \boldsymbol{\lambda} \rangle + o(\|\boldsymbol{\lambda}\|^2) \right)^2 \\
&\quad + o\left( \left( \langle M'(0), \boldsymbol{\lambda} \rangle + \frac{1}{2} \langle \boldsymbol{\lambda}, M''(0) \boldsymbol{\lambda} \rangle + o(\|\boldsymbol{\lambda}\|^2) \right)^2 \right) \\
&= \langle M'(0), \boldsymbol{\lambda} \rangle + \frac{1}{2} \left( \langle \boldsymbol{\lambda}, M''(0) \boldsymbol{\lambda} \rangle - \langle M'(0), \boldsymbol{\lambda} \rangle^2 \right) + o\left( \|\boldsymbol{\lambda}\|^2 \right) \\
&= \langle \boldsymbol{\mu}, \boldsymbol{\lambda} \rangle + \frac{1}{2} \boldsymbol{\lambda}^\top \Sigma \boldsymbol{\lambda} + o\left( \|\boldsymbol{\lambda}\|^2 \right),
\end{aligned}$$

where

$$\Sigma = M''(0) - M'(0)M'(0)^\top = \int (\mathbf{a} - \boldsymbol{\mu})(\mathbf{a} - \boldsymbol{\mu})^T u(\mathbf{a}) \mathrm{d}\mathbf{a}.$$

## B  Other perspectives to derive (3)

**A Maximum Likelihood Perspective.** The optimization problem in (3) can be derived using the maximum log likelihood method by treating the KL-divergence term as a regularizer. According to (2), the difference $(\boldsymbol{\mu} + \mathbf{z}) - \mathbf{h}$ follows a Gaussian distribution $\mathcal{N}(\mathbf{0}, \sigma^2 \mathbf{I})$. This implies the log likelihood function $\ell(\mathbf{z}) \propto -\frac{1}{2\sigma^2} \|(\boldsymbol{\mu} + \mathbf{z}) - \mathbf{h}\|^2$. Maximizing it with the KL-divergence term as a regularizer is the same as minimizing

$$\frac{1}{2\sigma^2} \|(\boldsymbol{\mu} + \mathbf{z}) - \mathbf{h}\|^2 + \eta \mathcal{K}(p, u), \tag{29}$$

where $\eta > 0$ controls the strength of the regularization. Substituting (1) into (29) followed by rearrangement yields

$$\min_{p} \frac{1}{2\eta\sigma^2} \left\| (\boldsymbol{\mu} + \mathbf{z}) - \int_{\mathbb{R}^d} \mathbf{a} p(\mathbf{a}) \, \mathrm{d}\mathbf{a} \right\|^2 + \mathcal{K}(p, u), \tag{30}$$

which is equivalent to (3) by setting $\alpha^{-1} = \eta\sigma^2$.

**A Bayesian perspective.** Given observed data and prior belief about the distribution of parameters, Bayesian inference allows us to update this distribution to reflect the new knowledge. Assume that the distribution $p$ is specified by parameters $\theta$. By considering $\boldsymbol{\mu} + \mathbf{z}$ as the observed data, we will show that picking the $p_\theta$ that minimizes (3) is the same as choosing the $\theta^*$ that maximizes the posterior density of $\theta$ given the observed data.

Let $\vartheta$ be the parameters of the preference distribution $u_\vartheta$ and suppose the prior distribution $f(\theta|\vartheta)$ of $\theta$ satisfies

$$f(\theta|\vartheta) \propto \exp\left( -\eta \mathcal{K}(p_\theta, u_\vartheta) \right), \tag{31}$$

where $\eta > 0$ is a hyper-parameter that controls the decaying speed of the probability density as $p_\theta$ deviates from $u_\vartheta$.

In (2), we have assumed that given $\theta$, $(\boldsymbol{\mu} + \mathbf{z}) - \mathbf{h}_\theta$ follows a spherical Gaussian distribution $\mathcal{N}(\mathbf{0}, \sigma^2 \mathbf{I})$, where $\mathbf{h}_\theta$ is the mean of $p_\theta$. Therefore, given its parameter $\theta$, the probability density function of $\boldsymbol{\mu} + \mathbf{z}$ is

$$f(\boldsymbol{\mu} + \mathbf{z}|\theta) = f(\boldsymbol{\mu} + \mathbf{z}|\mathbf{h}_\theta) \propto \exp\left( -\frac{1}{2\sigma^2} \|(\boldsymbol{\mu} + \mathbf{z}) - \mathbf{h}_\theta\|^2 \right). \tag{32}$$

Then the posterior distribution of $\theta$ satisfies

$$f(\theta|\boldsymbol{\mu} + \mathbf{z}, \vartheta) \propto f(\boldsymbol{\mu} + \mathbf{z}|\theta) \; f(\theta|\vartheta)$$

$$\propto \exp\left( -\frac{1}{2\sigma^2} \|(\boldsymbol{\mu} + \mathbf{z}) - \mathbf{h}_\theta\|^2 - \eta \mathcal{K}(p_\theta, u_\vartheta) \right).$$

Finding $\theta^*$ that maximizes the posterior $f(\theta|\boldsymbol{\mu} + \mathbf{z}, \vartheta)$ is the same as finding

$$p_\theta^* = \operatorname*{argmin}_{p_\theta} \left\{ \frac{1}{2\sigma^2} \|(\boldsymbol{\mu} + \mathbf{z}) - \mathbf{h}_\theta\|^2 + \eta \mathcal{K}(p_\theta, u_\vartheta) \right\}$$

$$= \operatorname*{argmin}_{p_\theta} \left\{ \frac{1}{2\eta\sigma^2} \|(\boldsymbol{\mu} + \mathbf{z}) - \mathbf{h}_\theta\| + \mathcal{K}(p_\theta, u_\vartheta) \right\},$$

which is equivalent to (3) by setting $\alpha^{-1} = \eta\sigma^2$.

# C   Extra experimental results

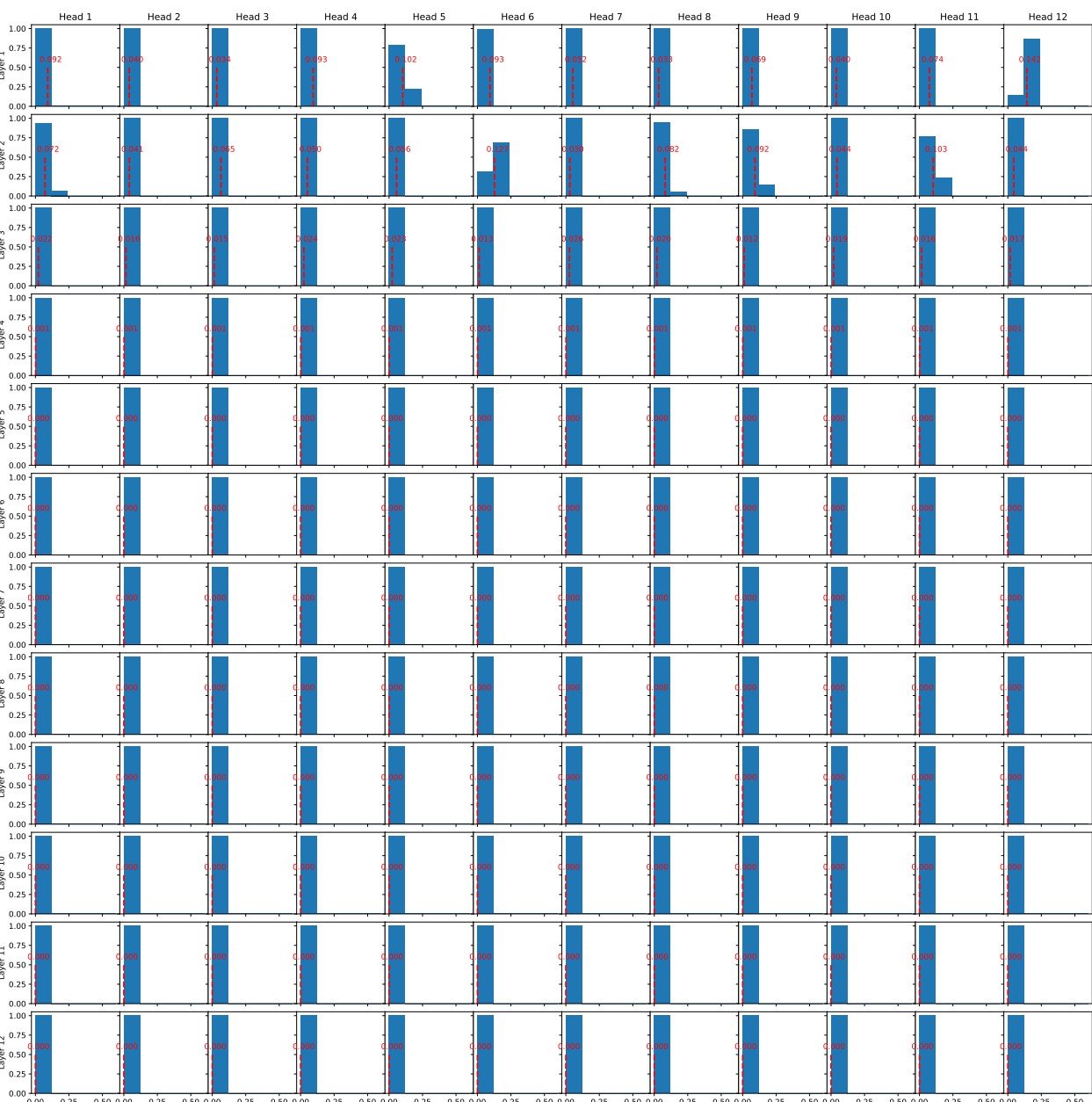

Figure 5: The distribution of relative errors $\frac{\|\boldsymbol{\lambda}^* - \alpha\mathbf{z}\|}{\|\boldsymbol{\lambda}^*\|}$ for the attention in BERT. The red vertical lines mark the average of the errors.

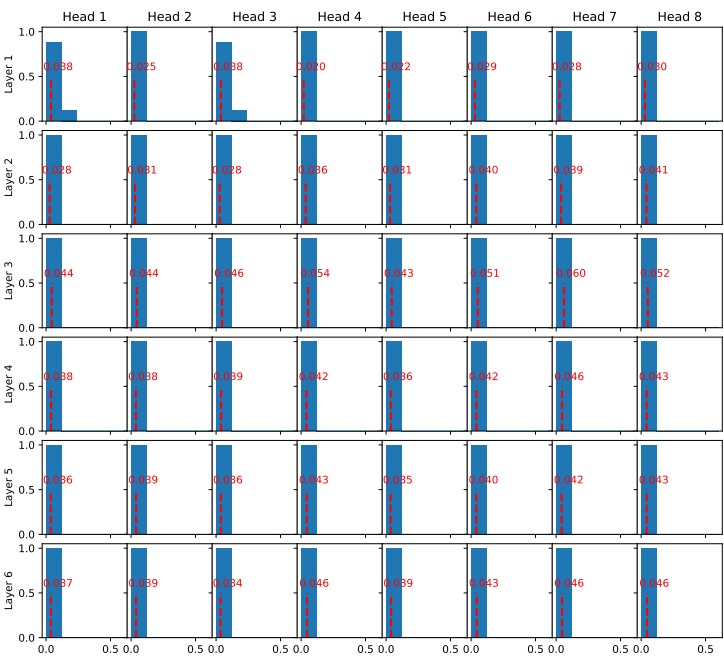

Figure 6: The distribution of relative errors $\frac{\|\boldsymbol{\lambda}^* - \alpha \mathbf{z}\|}{\|\boldsymbol{\lambda}^*\|}$ for the self-attention of the encoder in T5. The red vertical lines mark the average of the errors.

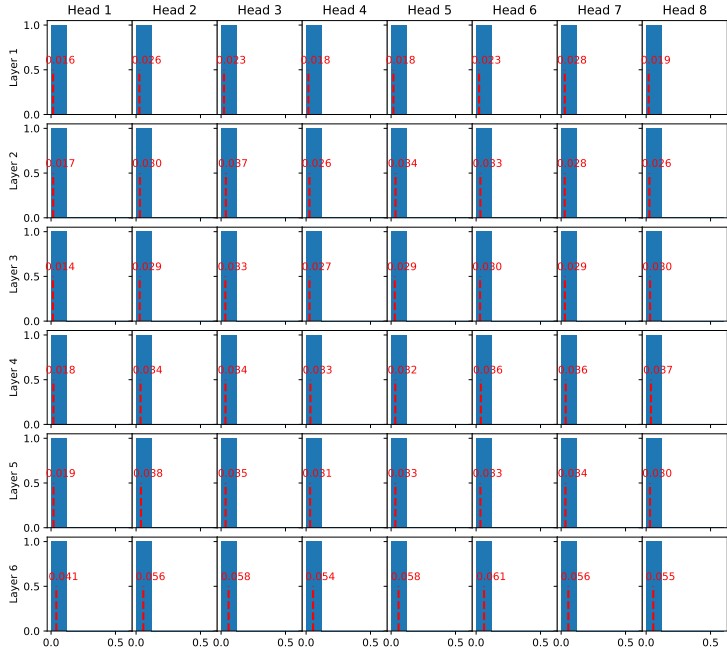

Figure 7: The distribution of relative errors $\frac{\|\boldsymbol{\lambda}^* - \alpha \mathbf{z}\|}{\|\boldsymbol{\lambda}^*\|}$ for the self-attention of the decoder in T5. The red vertical lines mark the average of the errors.

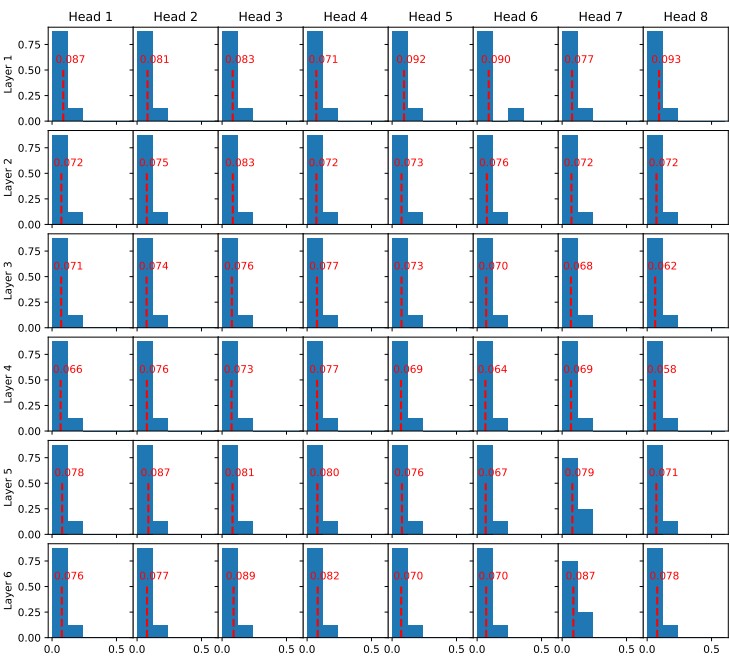

Figure 8: The distribution of relative errors $\frac{\|\boldsymbol{\lambda}^* - \alpha \mathbf{z}\|}{\|\boldsymbol{\lambda}^*\|}$ for the cross-attention in T5. The red vertical lines mark the average of the errors.

## D    Details on the derivation of OT-based attention

According to the discussion in Sec 9, we consider the optimization problem

$$p^* = \underset{p}{\operatorname{argmin}} \frac{\alpha}{2} \left\| (\tilde{\boldsymbol{\mu}} + \mathbf{z}) - \int_{\mathbb{R}^d} \mathbf{a} p(\mathbf{a}) \, \mathrm{d}\mathbf{a} \right\|^2 + \mathcal{W}_\gamma(p, u; M) \tag{33}$$

where $\tilde{\boldsymbol{\mu}}$ denotes the mean of the true preference distribution $\tilde{u}$ that minimizes $f(p) = \mathcal{W}_\gamma(p, u; M)$. We will show in Prop 1 that

$$\tilde{\boldsymbol{\mu}} = \sum_{\mathbf{t}, \mathbf{t}' \in \mathcal{A} \times \mathcal{A}} u(\mathbf{t}') \frac{\exp\left(-M(\mathbf{t}, \mathbf{t}')/\gamma\right)}{\sum_{\mathbf{t}'' \in \mathcal{A}} \exp\left(-M(\mathbf{t}'', \mathbf{t}')/\gamma\right)} \mathbf{t}. \tag{34}$$

Cuturi and Peyre Cuturi & Peyre (2016) proved that the Fenchel dual of $\mathcal{W}_\gamma(\mathbf{d}; r, M)$ is

$$\mathcal{W}_\gamma^*(p; u, M) = \gamma \left( H(u) + \sum_{\mathbf{t} \in \mathcal{A}} u(\mathbf{t}) \, \log \left[ \sum_{\mathbf{t}' \in \mathcal{A}} \exp\left( \gamma^{-1} \big( p(\mathbf{t}) - M(\mathbf{t}, \mathbf{t}') \big) \right) \right] \right) \tag{35}$$

for $p \in \mathbb{R}^N$; and, for $\mathbf{t} \in \mathcal{A}$

$$\left[ \nabla_p \mathcal{W}_\gamma^*(p; u, M) \right]_{\mathbf{t}} = \sum_{\mathbf{t}' \in \mathcal{A}} \frac{u(\mathbf{t}') \exp\left( \gamma^{-1} \big( p(\mathbf{t}) - M(\mathbf{t}, \mathbf{t}') \big) \right)}{\sum_{\mathbf{t}'' \in \mathcal{A}} \exp\left( \gamma^{-1} \big( p(\mathbf{t}'') - M(\mathbf{t}', \mathbf{t}'') \big) \right)}, \tag{36}$$

where $\left[ \nabla_p \mathcal{W}_\gamma^*(p; u, M) \right]_{\mathbf{t}}$ denote the entry in $\left[ \nabla_p \mathcal{W}_\gamma^*(p; u, M) \right]$ that is associated to template $\mathbf{t}$. By the Fenchel's duality theorem, we know that $p^*$ in (33) takes the form

$$p^*(\mathbf{t}) = \sum_{\mathbf{t}' \in \mathcal{A}} \frac{u(\mathbf{t}') \exp\left( \gamma^{-1} \big( \mathbf{t}^\top \lambda^* - M(\mathbf{t}, \mathbf{t}') \big) \right)}{\sum_{\mathbf{t}'' \in \mathcal{A}} \exp\left( \gamma^{-1} \big( (\mathbf{t}'')^\top \lambda^* - M(\mathbf{t}', \mathbf{t}'') \big) \right)}, \tag{37}$$

where

$$\lambda^* = \arg\max_{\lambda \in \mathbb{R}^d} \langle \tilde{\boldsymbol{\mu}} + \mathbf{z}, \lambda \rangle - \frac{1}{2\alpha} \|\lambda\|^2 - \mathcal{W}_\gamma^* \left([\mathbf{t}^T \lambda | \mathbf{t} \in \mathcal{A}]; u, M\right)$$

$$= \arg\max_{\lambda \in \mathbb{R}^d} \langle \tilde{\boldsymbol{\mu}} + \mathbf{z}, \lambda \rangle - \frac{1}{2\alpha} \|\lambda\|^2 - \gamma \sum_{\mathbf{t} \in \mathcal{A}} u(\mathbf{t}) \, \log \left[ \sum_{\mathbf{t}' \in \mathcal{A}} \exp\left( \frac{(\mathbf{t}')^\top \lambda - M(\mathbf{t}, \mathbf{t}')}{\gamma} \right) \right]. \tag{38}$$

**The true preference distribution.** The Fenchel dual perspective allows us to derive a closed-form expression for the minimizer of $f(p) = \mathcal{W}_\gamma(p, u; M)$, which we refer as the true preference distribution $\tilde{u}$ in the main text. We will also show that $\tilde{u}$ approaches to the preference $u$ as $\gamma \to 0$.

Notice that, by definition, $\tilde{u} \to p^*$ when $\alpha \to 0$ in (33). In this case, the optimization of $\lambda$ in (38) gets an infinite penalty on its L2 norm and thus $\|\lambda^*\|^2 = 0$. Therefore, we have

**Proposition 1.** $\mathcal{W}_\gamma(p; u, M)$ *has the minimizer* $\tilde{u}(\mathbf{t})$ *taking the form*

$$\tilde{u}(\mathbf{t}) = \sum_{\mathbf{t}' \in \mathcal{A}} u(\mathbf{t}') \frac{\exp\left(-M(\mathbf{t}, \mathbf{t}')/\gamma\right)}{\sum_{\mathbf{t}'' \in \mathcal{A}} \exp\left(-M(\mathbf{t}', \mathbf{t}'')/\gamma\right)}, \tag{39}$$

*for* $\mathbf{t} \in \mathcal{A}$. *Besides, its mean*

$$\tilde{\boldsymbol{\mu}} = \sum_{\mathbf{t}, \mathbf{t}' \in \mathcal{A} \times \mathcal{A}} u(\mathbf{t}') \, \frac{\exp\left(-M(\mathbf{t}, \mathbf{t}')/\gamma\right)}{\sum_{\mathbf{t}'' \in \mathcal{A}} \exp\left(-M(\mathbf{t}'', \mathbf{t}')/\gamma\right)} \mathbf{t}. \tag{40}$$

When $\gamma \to 0$, $\frac{\exp\left(-M(\mathbf{t}, \mathbf{t}')/\gamma\right)}{\sum_{\mathbf{t}'' \in \mathcal{A}} \exp(-M(\mathbf{t}', \mathbf{t}'')/\gamma)}$ approaches to 1 if $\mathbf{t} = \mathbf{t}'$ and 0 otherwise. Therefore, $\tilde{u}(\mathbf{t}) \to u(\mathbf{t})$ for all $\mathbf{t} \in \mathcal{A}$.

**The derivation of** (24)**.** Then we show how to derive (24) when $\alpha$ and $\frac{\alpha}{\gamma}$ are assumed small.

Within the summation term of (38), for a fixed $\mathbf{t}$

$$\log \left[ \sum_{\mathbf{t}' \in \mathcal{A}} \exp\left( \frac{(\mathbf{t}')^\top \lambda - M(\mathbf{t}, \mathbf{t}')}{\gamma} \right) \right] = \log \left[ \sum_{\mathbf{t}' \in \mathcal{A}} \exp\left( \frac{-M(\mathbf{t}, \mathbf{t}')}{\gamma} \right) \exp\left( \frac{(\mathbf{t}')^\top \lambda}{\gamma} \right) \right]$$

$$= \log \left[ \sum_{\mathbf{t}' \in \mathcal{A}} q_{\mathbf{t}}(\mathbf{t}') Z(\mathbf{t}) \, \exp\left( \frac{(\mathbf{t}')^\top \lambda}{\gamma} \right) \right] = \log \left[ \sum_{\mathbf{t}' \in \mathcal{A}} q_{\mathbf{t}}(\mathbf{t}') \, \exp\left( \frac{(\mathbf{t}')^\top \lambda}{\gamma} \right) \right] + \log Z(\mathbf{t})$$

$$= \log \mathbf{M}_{\mathbf{t}}(\lambda/\gamma) + \log Z(\mathbf{t}), \tag{41}$$

where

$$q_{\mathbf{t}}(\mathbf{t}') = \exp\left( \frac{-M(\mathbf{t}, \mathbf{t}')}{\gamma} \right) \Big/ Z(\mathbf{t}),$$

$$Z(\mathbf{t}) = \sum_{\mathbf{t}' \in \mathcal{A}} \exp\left( \frac{-M(\mathbf{t}, \mathbf{t}')}{\gamma} \right),$$

and $\mathcal{M}_{\mathbf{t}}$ denotes the MGF of $q_{\mathbf{t}}$.

Note that $\log \mathcal{M}_{\mathbf{t}}(\lambda/\gamma)$ is called the cumulant of $q_{\mathbf{t}}$ and has the expansion

$$\log \mathcal{M}_{\mathbf{t}}(\lambda/\gamma) = \boldsymbol{\mu}_{\mathbf{t}}^\top (\lambda/\gamma) + \frac{1}{2} \, (\lambda/\gamma)^\top \, \Sigma_{\mathbf{t}} \, (\lambda/\gamma) + \mathcal{O}(\|\lambda/\gamma\|^3), \tag{42}$$

where

$$\boldsymbol{\mu}_{\mathbf{t}} = \sum_{\mathbf{t}' \in \mathcal{A}} q_{\mathbf{t}}(\mathbf{t}') \, \mathbf{t}' \tag{43}$$

and

$$\Sigma_{\mathbf{t}} = \sum_{\mathbf{t} \in \mathcal{A}} q_{\mathbf{t}}(\mathbf{t}') \ (\mathbf{t}' - \mu_{\mathbf{t}})(\mathbf{t}' - \mu_{\mathbf{t}})^{\top} \tag{44}$$

respectively denote the mean and the variance-covariance matrix of $q_{\mathbf{t}}$.

Substituting (41) and (42) into (38) yields

$$\begin{aligned}
\lambda^* = \arg\max_{\lambda \in \mathbb{R}^d} &\langle \tilde{\boldsymbol{\mu}} + \mathbf{z}, \lambda \rangle - \frac{1}{2\alpha} \|\lambda\|^2 \\
&- \gamma \left[ \left( \sum_{\mathbf{t} \in \mathcal{A}} u(\mathbf{t}) \boldsymbol{\mu}_{\mathbf{t}} \right)^{\top} (\lambda/\gamma) + \frac{1}{2} \sum_{\mathbf{t} \in \mathcal{A}} u(\mathbf{t}) \left( (\lambda/\gamma)^{\top} \Sigma_{\mathbf{t}} (\lambda/\gamma) \right) + \mathcal{O}(\|\lambda/\gamma\|^3) + \sum_{\mathbf{t} \in \mathcal{A}} u(\mathbf{t}) \log Z(\mathbf{t}) \right] \\
= \arg\max_{\lambda \in \mathbb{R}^d} &\langle \tilde{\boldsymbol{\mu}} + \mathbf{z}, \lambda \rangle - \frac{1}{2\alpha} \|\lambda\|^2 \\
&- \gamma \left[ (\sum_{\mathbf{t} \in \mathcal{A}} u(\mathbf{t}) \boldsymbol{\mu}_{\mathbf{t}})^{\top} (\lambda/\gamma) + \frac{1}{2} \sum_{\mathbf{t} \in \mathcal{A}} u(\mathbf{t}) \left( (\lambda/\gamma)^T \Sigma_{\mathbf{t}} (\lambda/\gamma) \right) + \mathcal{O}(\|\lambda/\gamma\|^3) \right] \\
= \arg\max_{\lambda \in \mathbb{R}^d} &\langle \tilde{\boldsymbol{\mu}} + \mathbf{z}, \lambda \rangle - \frac{1}{2\alpha} \|\lambda\|^2 \\
&- \left[ \left( \sum_i u(\mathbf{t}) \boldsymbol{\mu}_{\mathbf{t}} \right)^{\top} \lambda + \frac{1}{2\gamma} \sum_{\mathbf{t} \in \mathcal{A}} u(\mathbf{t}) \left( \lambda^{\top} \Sigma_{\mathbf{t}} \lambda \right) + \gamma \mathcal{O}(\|\lambda/\gamma\|^3) \right].
\end{aligned}$$

When $\alpha$ is assumed to be small, the optimization of $\lambda$ gets a large penalty on its L2 norm and thus, $\|\lambda^*\|^2$ is close to zero. So we have

$$\begin{aligned}
\lambda^* \approx \arg\max_{\lambda \in \mathbb{R}^d} &\langle \tilde{\boldsymbol{\mu}} + \mathbf{z}, \lambda \rangle - \frac{1}{2\alpha} \|\lambda\|^2 \\
&- \left[ \left( \sum_{\mathbf{t} \in \mathcal{A}} u(\mathbf{t}) \boldsymbol{\mu}_{\mathbf{t}} \right)^{\top} \lambda + \frac{1}{2\gamma} \sum_{\mathbf{t} \in \mathcal{A}} u(\mathbf{t}) \left( \lambda^{\top} \Sigma_{\mathbf{t}} \lambda \right) \right]
\end{aligned}$$

Taking the derivative in terms of $\lambda$ and setting it to zero yields

$$(\tilde{\boldsymbol{\mu}} + \mathbf{z}) - \frac{1}{\alpha} \lambda^* - \sum_{\mathbf{t} \in \mathcal{A}} u(\mathbf{t}) \boldsymbol{\mu}_{\mathbf{t}} - \frac{1}{\gamma} \sum_{\mathbf{t} \in \mathcal{A}} u(\mathbf{t}) \Sigma_{\mathbf{t}} \lambda^* = 0.$$

As

$$\sum_{\mathbf{t} \in \mathcal{A}} u(\mathbf{t}) \boldsymbol{\mu}_{\mathbf{t}} = \sum_{i=1}^{N} u(\mathbf{t}) \sum_{\mathbf{t}' \in \mathcal{A}} q_{\mathbf{t}}(\mathbf{t}') \mathbf{t}' = \sum_{\mathbf{t}, \mathbf{t}' \in \mathcal{A} \times \mathcal{A}} u(\mathbf{t}) \frac{\exp\left(-M(\mathbf{t}, \mathbf{t}')/\gamma\right)}{\sum_{\mathbf{t}'' \in \mathcal{A}} \exp\left(-M(\mathbf{t}, \mathbf{t}'')/\gamma\right)} \mathbf{t}' = \tilde{\boldsymbol{\mu}}, \tag{45}$$

we also have

$$\mathbf{z} - \left( \frac{1}{\alpha} I_d + \frac{1}{\gamma} \sum_{\mathbf{t} \in \mathcal{A}} u(\mathbf{t}) \Sigma_{\mathbf{t}} \right) \lambda^* = 0.$$

That is,

$$\begin{aligned}
\lambda^* &= \left( \frac{1}{\alpha} I_d + \frac{1}{\gamma} \sum_{\mathbf{t} \in \mathcal{A}} u(\mathbf{t}) \Sigma_{\mathbf{t}} \right)^{-1} \mathbf{z} \\
&= \left( I_d + \frac{\alpha}{\gamma} \sum_{\mathbf{t} \in \mathcal{A}} u(\mathbf{t}) \Sigma_{\mathbf{t}} \right)^{-1} (\alpha \mathbf{z}).
\end{aligned}$$

When $\frac{\alpha}{\gamma}$ is small, the expression becomes simply

$$\lambda^* = \alpha \mathbf{z}.$$

Plugging it into (37), we get

$$p^*(\mathbf{t}) = \sum_{\mathbf{t}' \in \mathcal{A}} \frac{u(\mathbf{t}') \exp\left(\gamma^{-1}\big(\alpha \mathbf{t}^\top \mathbf{z} - M(\mathbf{t}, \mathbf{t}')\big)\right)}{\sum_{\mathbf{t}'' \in \mathcal{A}} \exp\left(\gamma^{-1}\big(\alpha (\mathbf{t}'')^\top \mathbf{z} - M(\mathbf{t}', \mathbf{t}'')\big)\right)}, \tag{46}$$

which is (24).

