# OpenReview forum: "Attention as Inference via Fenchel Duality"
_TMLR — Rejected by TMLR_

### Review · Reviewer_9ZFK · 2022-11-10

**Summary Of Contributions:**

The paper tackles the task of providing a theoretical model that can help understanding and interpreting the attention mechanisms that spawned many of the state-of-the-art architectures in many applications. Looking at different practical examples where attention-based models have shown interesting improvements, namely translation, image captioning and filling the blank (as used in training the BERT model), they show that the attention block can be seen as a solver for a certain type of optimization problem, that can be briefly described as follows:
* The main focus is to estimate the mean vector $h^{(k)}$ of an unknown probability distribution $p^{(k)}$ on $\mathbb{R}^d$.
* In order to do that, we are given another distribution $u^{(k)}$, called the preference distribution, that can be the output of a network module $f_{\text{pref}}^{(k)}$. This distribution gives non zero masses $u_1^{(k)}, \dots, u_M^{(k)}$ for some template vectors $t_1^{(k)}, \dots, t_M^{(k)}$. These template can be for example the source sentence word embeddings, that are obtained by 2 LSTM layers as in the model for translation in (Bahadanau et al., 2014). The probability masses in this model correspond to the output of the position encoding function.
* We are also given another source of information $z^{(k)}$, called evidence, that can also be generated by another neural module, and that is assumed to give a noisy version of the shift from the true distribution, meaning that is $\mu^{(k)}$ is the mean of $u^{(k)}$, then $z^{(k)} = h^{(k)} - \mu^{(k)} + \epsilon$ where $\epsilon$ is a spherical Gaussian noise. In the model stated earlier, this can correspond to the encoding of the previous words that is operated by another LSTM layer.

This setting can be translated into the following optimization problem: $\min_p \frac{\alpha}{2} ||(\mu + z) - \int_{\mathbb{R}^d} xp(x)dx||^2 + KL(p,u)$, where $KL$ designates the KL-divergence. Inspired by (Rioux et al. 2020), the authors first formulate the Fenchel dual of this problem, and express an estimate of the target mean $h$ based on its optimizer. They however suggest that this optimizer cannot be expressed exactly, but can be approximated.

This approximation leads to an estimate of the mean $h$ that gives a generalized form of the attention block. A special case of this obtained expression is the popular dot-product attention. The authors also suggest that the successful T5 architecture emulates this generalized attention implicitly. They validate their theory by comparing the approximate optimizer of the problem they propose to the output of the attention block in BERT.



**Audience:**

Yes

**Broader Impact Concerns:**

The attention mechanisms are employed in many state-of-the-art models that have a significant social impact. It would be interesting to consider if the current perspective can enlighten some properties of these modules from a bias/fairness and other ethical perspectives.

**Claims And Evidence:**

No

**Requested Changes:**

Critical changes:
* Regarding clarity, while I appreciated the use of practical examples, the paper structure makes it a bit confusing and quite hard to follow. Here are some suggestions to make the paper clearer:
    - Rather than starting from the motivating example without prior grounding, it would be much clearer to start by describing the practical examples, show that they can be unified and abstracted in a similar way, which leads to the design problem and the introduction of the optimization problem, after which the motivating example can be introduced as a guide/tool to solve the optimization problem. The authors describe it as it is "a seemingly unrelated example". It is much less so when introduced in the same place.
   - For a better understanding of the different components in the model the authors propose (preference/evidence modules), and for the self sufficiency of the the work, it would be better to first describe the models in the practical models as they appear in the literature, before translating them in the framework introduced in this paper.

* Regarding the experimental experiments, it seems insufficient to show that this hold for a particular problem, especially a problem that is used in the model definition. It would be interesting to validate this not only for the settings described in the practical examples, but also for other attention based models, to show that this interpretation generalizes beyond the problems from which it is derived. It would be interesting also to show that the observations hold independently not only of the definition of the preference and evidence modules (i.e. different architectures) but also of the input distribution. Finally the authors emphasize that T5 implicity operates this generalized attention formulation, it would be interesting to validate this empirically. The authors do say that they tested with other settings, but they don't report the results.

Non-critical changes/questions:
* Can the author comment on the possibility of using their results to improve the attention based models. For example, is it possible/practical to replace the attention block by the dual problem optimizer introduced in this paper? How expensive will it be? Would it hurt/help the performance?
* Would it be possible build on this result to get insights on the stability/robustness/performance implication of these modules?
* The solution of the optimization problem is based on an approximation. Can the author comment on the error incurred by this approximation, especially when accumulated other multiple layers?

**Strengths And Weaknesses:**

Strengths:
* The problem tackled in the paper is important and of significance to the community.
* The approach that the authors take is sound and makes sense. They ground it in relevant related works, although I cannot comment on its coverage of the existent work, not being expert on the field.
* I didn't detect a significant flaw in the derivation.
* The theory is also related to practical problems that makes it easier to ground in real applications.

Weaknesses:
* The clarity of the paper could be improved (see suggested changes below)
* The empirical validation is relatively weak and seems not sufficient to validate the theory.

---

> ### Author Response · Authors · 2022-12-07
> **Clarifications (Part 1)**
>
> Thank you for reviewing our paper and providing us with constructive feedback. We are glad that you found our theoretical work interesting. In the discussion below, we seek to address your questions and explain the corresponding changes made in our revised submission.
>
>
> **Q1:** Regarding clarity, while I appreciated the use of practical examples, the paper structure makes it a bit confusing and quite hard to follow. Here are some suggestions to make the paper clearer:
>
> - Rather than starting from the motivating example without prior grounding, it would be much clearer to start by describing the practical examples, show that they can be unified and abstracted in a similar way, which leads to the design problem and the introduction of the optimization problem, after which the motivating example can be introduced as a guide/tool to solve the optimization problem. The authors describe it as it is "a seemingly unrelated example". It is much less so when introduced in the same place.
> - For a better understanding of the different components in the model the authors propose (preference/evidence modules), and for the self sufficiency of the the work, it would be better to first describe the models in the practical models as they appear in the literature, before translating them in the framework introduced in this paper.
>
> **A:** *We think this is a good suggestion and have made the corresponding changes in our revised submission.*
>
> **Q2:** Regarding the experimental experiments, it seems insufficient to show that this hold for a particular problem, especially a problem that is used in the model definition. It would be interesting to validate this not only for the settings described in the practical examples, but also for other attention based models, to show that this interpretation generalizes beyond the problems from which it is derived. It would be interesting also to show that the observations hold independently not only of the definition of the preference and evidence modules (i.e. different architectures) but also of the input distribution. Finally the authors emphasize that T5 implicity operates this generalized attention formulation, it would be interesting to validate this empirically. The authors do say that they tested with other settings, but they don't report the results.
>
> **A:** *We have provided extra empirical results on T5 model in Sec 8 and Appx C. The results are consistent with what we have obtained from BERT. Considering that T5 structures were not covered when we derived the framework, we believe this addresses the first and the last points.*
>
> *Regarding "show the observations hold independently not only of the definition of the preference and evidence modules (i.e. different architectures) but also of the input distribution," we want to note that it is a common practice to preprocess the input data by normalizations and neural networks before feeding them into attention structures. Our work does not suggest that our formulation also applies to the input distribution; instead, we believe other modules of the model collaborate with the attention layer to make the approximation presented in our work valid.*
>
> **Q3:** Can the author comment on the possibility of using their results to improve the attention based models. For example, is it possible/practical to replace the attention block by the dual problem optimizer introduced in this paper? How expensive will it be? Would it hurt/help the performance?
>
> **A:** *We are investigating this problem at the moment. Generally speaking, using the dual problem optimizer in the forward pass does not significantly slow down the computation efficiency; however, it will slow down the backward pass and make the model optimization unstable. This is somewhat similar to a recurrent network that has infinitely many layers. We plan to alleviate this problem using some new results reported in the deep equilibrium models; however, how much computation efficiency we can improve is still largely unknown at the moment.*
>
> **Q4:** Would it be possible build on this result to get insights on the stability/robustness/performance implication of these modules?
>
> **A:** *Our theoretical work demonstrates a connection between the attention structure and a family of convex optimization problems. We believe this helps us to understand the attention mechanism from a convex optimization perspective. Considering that there are lots of work on the stability and robustness of solutions in the convex optimization field, we believe our work allows us to leverage those insights to characterize attention structure's behaviours.*

---

> > ### Author Response · Authors · 2022-12-07
> > **Clarifications (Part 2)**
> >
> > **Q5:** The solution of the optimization problem is based on an approximation. Can the author comment on the error incurred by this approximation, especially when accumulated other multiple layers?
> >
> > **A:** *Lots of empirical research in Deep Learning suggests that neural networks are quite robust to errors/noises and may still perform well even if the models are not implemented correctly. Although our work suggests that the attention structure cannot solve the optimization problem perfectly, the approximation error can be handled by other components and will not accumulate to cause a catastrophic performance drop. On the other hand, if a new structure can be found that solves the optimization problem more accurately, the model's performance is expected to improve.*
> >
> > **Q6:** The attention mechanisms are employed in many state-of-the-art models that have a significant social impact. It would be interesting to consider if the current perspective can enlighten some properties of these modules from a bias/fairness and other ethical perspectives.
> >
> > **A:** *We have added a paragraph at the end of the main text to discuss our work's broader impacts.*

---

### Review · Reviewer_Uncs · 2022-11-11

**Summary Of Contributions:**

The authors demonstrate an equivalence between an attention layer and a regularized optimization problem. In particular, they demonstrate that the output of a single attention layer corresponds to an estimator of the mean of a distribution given a noisy observation and regularized by a prior distribution.

**Audience:**

Yes

**Broader Impact Concerns:**

None.

**Claims And Evidence:**

No

**Requested Changes:**

## Critical changes

- Fix (or remove) the Bayesian interpretation.
- Fix language about placing a prior on `p`.
- Tone down the claims of "theoretical insight".

Moreover, it would be very useful to demonstrate how this connection can lead to better design of attention mechanisms. You mention that a "future paper" designs new attention mechanisms based on the insight of this work. I would suggest including some of those results in this paper. Demonstrating \*why\* this interpretation is useful would make this paper far more relevant and interesting to the community at large.

**Strengths And Weaknesses:**

## Strengths

- The paper is clearly written
- The authors offer detailed descriptions of transformers and the tasks that they are applied to
- The math is (mostly) sound, though with a few critical exceptions.

## Weaknesses

### Claims of "theoretical insight" are too strong

This paper motivates itself as a "principled justification for the design of attention modules." However, I would argue that the paper comes up far short of this. The primary point of the paper is demonstrating a connection between attention layers and an optimization problem. At the same time, I would argue that this does not constitute a "justification." (See next section.)

### Incorrect mathematical language, and inaccuracies with the Bayesian perspective.

- Section 3: "Assume a probability distribution p on $\mathbb R^d$ has a spherical Gaussian prior." A distribution cannot have a prior.
- Equation 11: A likelihood should be conditioned on a parameter, not a distribution. In other words, the equation $\Pr(\boldsymbol \mu + \mathbf z \mid p)$ is not a valid mathematical idea.
- Equation 12: Again, a distribution cannot have a prior.

As it stands, because of these mathematical errors, the Bayesian perspective is inaccurate.

### Section 5 is well known

The purpose of Section 5 is to demonstrate that the optimization problem can be interpreted under regularized maximum likelihood/maximum entropy/maximum a posteriori perspectives. This section is - to a large degree - well known and trivial. The connection between a regularized maximum likelihood objective and a maximum a posteriori objective is common knowledge, as is the connection between a regularized objective (Eq. 9) and a constrained optimization problem (Eq. 10). I believe these connections will be obvious to anyone with a basic machine learning background, and should not take up a full page of a research publication.

### Small notes

- The parameter $\nu$ in Section 5 is superfluous. From the ML and Bayesian perspectives, you can recover the optimization objective by setting `$\nu=1$` and $\alpha^{-1} = \sigma^2$.

---

> ### Author Response · Authors · 2022-12-07
> **Clarifications.**
>
> We thank you for your time and comments.  We have given the following responses/explanations to clarify aspects of our paper and made corresponding changes in the revised submission.
>
> **Q1:** *This paper motivates itself as a "principled justification for the design of attention modules." However, I would argue that the paper comes up far short of this. The primary point of the paper is demonstrating a connection between attention layers and an optimization problem. At the same time, I would argue that this does not constitute a "justification."*
>
> **A:** *We believe our work does provide some justification for the design of attention. Specifically, there is a general assumption that because attention in humans narrows the search space, a similar phenomenon is at play in transformers. In this paper, we have shown that the mechanism corresponds to proposing a preference distribution over the templates, followed by adjusting it using a noisy mean shift estimation. We have provided extra details in Conclusion to clarify our statements. Additionally, we have added Sec 9 in our revised submission to show how our framework helps develop new attention structures, which we believe can further corroborate our claim.*
>
> **Q2:** Incorrect mathematical language, and inaccuracies with the Bayesian perspective.
>  - Section 3: "Assume a probability distribution p on  has a spherical Gaussian prior." A distribution cannot have a prior.
>
>  **A:** We were using this shorthand to avoid excessive notation but we understand it is not common and we have switched to more standard language. As part of moving the motivating example (see answers above) it now appears in Section 5 and we have improved its description.
>
>  - Equation 11: A likelihood should be conditioned on a parameter, not a distribution. In other words, the equation  is not a valid mathematical idea.
>  - Equation 12: Again, a distribution cannot have a prior.
>
>  **Note:** *We have moved this part to Appx B.*
>
> **A:** We agree that our original formulation is not very formal. We slightly abused the notations to simplify the expressions, using the same symbol to refer to a distribution, its probability density function (pdf), and - as we warned in the text - "p as a model parameter." We agree, however, this is somewhat far from mathematical custom, and we have made necessary changes to make everything precise.
>
>
>
> **Q3:**  Section 5 is well known
>
> **A:** *We provided the three perspectives in the main text as we thought they would help readability. We are aware that Section 5 may get some of the potential audience bored, and we have moved the Maximum Likelihood Perspective and the Bayesian perspective to Appx B*.
>
> **Q4:** The parameter $\nu$ in Section 5 is superfluous. From the ML and Bayesian perspectives, you can recover the optimization objective by setting $\nu=1$ and $\alpha^{-1} =\sigma^2$.
>
> **A:** *We believe you were referring to $\eta$. We agree that $\eta$ is unnecessary as it can be absorbed into $\sigma^2$. We keep it here because we want to keep an explicit parameter to control the relative strength of the two terms (just like the $\alpha$ in (3))*.
>
> **Q5:** Moreover, it would be very useful to demonstrate how this connection can lead to better design of attention mechanisms. You mention that a "future paper" designs new attention mechanisms based on the insight of this work. I would suggest including some of those results in this paper. Demonstrating *why* this interpretation is useful would make this paper far more relevant and interesting to the community at large.
>
> **A:** *We have added Sec 9 in the revision to show how our framework facilitates the development of new attention structures.*

---

> > ### Comment · Reviewer_Uncs · 2022-12-09
> > **Response to updates**
> >
> > Thank you for addressing my concerns.
> >
> > I appreciate the new Section 9 and its discussion about potential new attention structures. I believe that this paper could be even stronger if this section was expanded - though such an expansion would necessitate another round of reviews. Nevertheless, after you have addressed my concerns, this paper satisfies the correctness and audience requirements of TMLR, so I will recommend acceptance of this paper.

---

### Review · Reviewer_JBG5 · 2022-11-23

**Summary Of Contributions:**

This paper analyses a specific entropy regularized optimization problem, brings Fenchel duality to bear to solve it, and notices that the solution is the Boltzmann distribution, i.e. a softmax over a function of some given states corresponding to their "energy". This specific shape of distribution (a softmax over a function of discrete states) prompts the authors to draw an analogy with the self-attention operation in transformer architectures, and propose that the solution to their optimization problem should be thought of as a form of generalized attention. The paper then claims that the success of transformer architectures like T5 is due to them implicitly adopting such a form of generalized attention.

**Audience:**

No

**Broader Impact Concerns:**

There aren't any major ethical implications of the work.

**Claims And Evidence:**

No

**Requested Changes:**

The paper would definitely benefit from restructuring the presentation to make the main message less opaque. As of now, the paper comes across as unnecessarily convoluted, with the result that the main claims appear as overblown and unsupported by the technical work. It is however unclear whether clarifying the presentation, the notation, and recalibrating the claims of the paper would be enough to grant it enough strength as a publication in the journal.

**Strengths And Weaknesses:**

Presentation: The paper's presentation and structure are unnecessarily convoluted, to the point of making the message opaque. The paper unnecessarily goes back and forth on the optimization problem, first introducing it as a motivating example, later as the central optimization problem. The notation is arguably unwieldy and contributes to making a simple optimization problem and the reading of the paper more laborious than needed.
Technical soundness: Putting Fenchel duality front and center in the presentation seems odd and incurs the risk of making the paper seem more unapproachable and technically involved than it actually is. In the end, as the paper itself also points out, the solution to the optimization problem can be obtained simply using maximum log likelihood or constrained maximum entropy. The heavy focus on Fenchel duality, which is just another technique to solve the problem, is then arguably arbitrary, with the disadvantage of again contributing to the possible opaqueness of the paper.
The claim that T5 is as successful as it is because it is implicitly adopting a generalized attention as the one specified in this paper seems oddly specific and difficult to justify. First of all, there is nothing in the "generalized attention" formulation in this paper that specifically corresponds to relative positional encodings, which are an architectural construct that makes sense in the domain-specific situation where the proximity of tokens in the 1D sequence of tokens in NLP is meaningful. On the other hand, the generalized attention expressed in the paper is too generic to prescribe domain-specific architectural features like relative positional encodings. In addition, the success of T5 is presumably not solely due to relative positional encodings.

---

> ### Author Response · Authors · 2022-12-07
> **Clarifications.**
>
> Thank you for taking the time to review our paper. Our responses seek to clarify our results and what we have done in the revised version to avoid potential confusion.
>
> **Q1:** The paper's presentation and structure are unnecessarily convoluted, to the point of making the message opaque. The paper unnecessarily goes back and forth on the optimization problem, first introducing it as a motivating example, later as the central optimization problem. The notation is arguably unwieldy and contributes to making a simple optimization problem and the reading of the paper more laborious than needed.
>
> **A:** *We are aware that the location of the motivating example and the way to introduce the optimization problem may cause some confusion. We have made necessary adjustments according to the suggestions of 9ZFK (Q1)*.
>
>
> **Q2:**  Putting Fenchel duality front and center in the presentation seems odd and incurs the risk of making the paper seem more unapproachable and technically involved than it actually is. In the end, as the paper itself also points out, the solution to the optimization problem can be obtained simply using maximum log likelihood or constrained maximum entropy. The heavy focus on Fenchel duality, which is just another technique to solve the problem, is then arguably arbitrary, with the disadvantage of again contributing to the possible opaqueness of the paper.
>
> **A:** *We want to note that the three perspectives presented in Sec 4 (and Appx B) are used to help readers understand our proposed optimization problem; however, they are irrelevant to solving the problem itself. To solve the optimization problem, we have to instead solve its Fenchel dual problem and then convert the found solution back to the one of the original problem. We then show that this solution essentially adopts a generalized attention structure. We have adjusted our paper and rewritten the motivating example section to better explain the logic of our writing and the derivation of our results.*
>
>
> **Q3:** The claim that T5 is as successful as it is because it is implicitly adopting a generalized attention as the one specified in this paper seems oddly specific and difficult to justify. First of all, there is nothing in the "generalized attention" formulation in this paper that specifically corresponds to relative positional encodings, which are an architectural construct that makes sense in the domain-specific situation where the proximity of tokens in the 1D sequence of tokens in NLP is meaningful. On the other hand, the generalized attention expressed in the paper is too generic to prescribe domain-specific architectural features like relative positional encodings. In addition, the success of T5 is presumably not solely due to relative positional encodings.
>
> **A:** *We would like to note that we do not try to claim the T5 attention succeeds because it implicitly adopts our generalized attention structure. Instead, all we have claimed is that "T5 transformer can be seen as a natural realization of the generalized attention, where the preference weights $u$ unifies the concepts of word masks and T5's positional encoding functions". As a result, the usefulness and the validity of our framework are well-supported by the state-of-the-art performance of T5. In Sec 7, we mathematically show why T5 can be seen as a special case of the generalized attention defined in (17); therefore, we believe our claim is well-supported. We have rephrased the related statement in the revised version to avoid potential confusion.*

---

### Decision · Action_Editors · 2022-12-20

**Recommendation:** Reject

**Comment:**

There was not a consensus on this paper. Reviewers were mainly initially concerned about the presentation and correctness. The paper provided motivating examples first before presenting the framework, which reviewers agreed was confusing; the updated draft has changed this. There were many issues with correctness and notation, some of which have been corrected. One reviewer specifically requested additional insight on how the proposed framework could be used to design improved attention mechanisms. A small section with additional discussion of this possibility was added, but it is probably insufficient to be considered a contribution on its own. Reviewers also requested additional experiments on T5, which were added to the appendix, but would likely make the paper stronger if they were in the main text. Given the various concerns of the reviewers, some of which were addressed during the rebuttal (but requiring substantial changes to the paper), the submission should undergo another significant revision before being resubmitted.

**Audience:**

There are likely some members of the TMLR audience who would be interested to hear this perspective.

**Claims And Evidence:**

Overall, the papers theoretical claims are justified. Reviewers pointed out a few issues with the notation and definitions, which seem to have been fixed in the updated version. Some reviewers argued that the theoretical claims were trivial. The paper also includes some experiments to provide empirical evidence of their perspective. The authors included additional experiments on T5 (which is a real-world example of a model that accidentally follows the framework proposed in the paper) in the appendix, though these should probably go in the main text given that the paper relies on T5 as indirect evidence in favor of the framework.